# Optimal Dynamic Regret by Transformers for Non-Stationary Reinforcement Learning

**Baiyuan Chen**
The University of Tokyo
chenbaiyuan75@g.ecc.u-tokyo.ac.jp

**Shinji Ito**
The University of Tokyo / RIKEN AIP
shinji@mist.i.u-tokyo.ac.jp

**Masaaki Imaizumi**
The University of Tokyo / RIKEN AIP
imaizumi@g.ecc.u-tokyo.ac.jp

## Abstract

Transformers have demonstrated exceptional performance across a wide range of domains. While their ability to perform reinforcement learning in-context has been established both theoretically and empirically, their behavior in non-stationary environments remains less understood. In this study, we address this gap by showing that transformers can achieve nearly optimal dynamic regret bounds in non-stationary settings. We prove that transformers are capable of approximating strategies used to handle non-stationary environments and can learn the approximator in the in-context learning setup. Our experiments further show that transformers can match or even outperform existing expert algorithms in such environments.

## 1 Introduction

Transformers have emerged as a powerful class of sequence models with remarkable expressive capabilities. Originally popularized in the context of natural language processing, they leverage self-attention mechanisms to in-context learn new tasks without any parameter updates (Vaswani et al., 2017; Liu et al., 2021; Dosovitskiy et al., 2021; Yun et al., 2019; Dong et al., 2018). In other words, a large transformer model can be given a prompt consisting of example input-output pairs for an unseen task and subsequently produce correct outputs for new queries of that task, purely by processing the sequence of examples and queries (Lee et al., 2022; Laskin et al., 2023; Yang et al., 2023; Lin et al., 2024). This ability to dynamically adapt via context rather than gradient-based fine-tuning has spurred extensive interest in understanding the theoretical expressivity of transformers and how they might *learn algorithms* internally during training.

Recent theoretical work has begun to analyze the various aspects of transformers. On the expressiveness front, transformers have been shown to be universal or near-universal function approximators under various conditions (Yun et al., 2020). Beyond mere approximation of static functions, researchers have demonstrated that transformers can encode and execute entire computational procedures. For example, by carefully setting a transformer's weights, one can make its forward pass simulate iterative algorithms such as gradient descent and many others (Bai et al., 2023; Cheng et al., 2024; Huang et al., 2025).

Studies have shown that transformers can approximate various reinforcement learning (RL) algorithms. The question here is whether a transformer can be trained to act as an RL algorithm when presented with an interaction history as context. Lin et al. (2024) shows that a sufficiently large transformer model, after being pre-trained on offline interaction sequences, can approximate near-

39th Conference on Neural Information Processing Systems (NeurIPS 2025).

| Study | Lin et al. (2024) (LinUCB) | Lin et al. (2024) (TS) | Ours |
|---|---|---|---|
| Dynamic Regret | $O(T)$ | $O(\Delta^{1/4}T^{3/4})$ | $O(\min\{\sqrt{JT}, \Delta^{1/3}T^{2/3} + \sqrt{T}\})$ |

Table 1: Upper bounds on dynamic regret of transformers in the non-stationary setup. $T$ is the time budget, and $\Delta$ and $J$ represent the amount and the number of changes in the environment, respectively. LinUCB suffers from $O(T)$ dynamic regret in non-stationary environments. The rate of Thompson Sampling (TS) is from Kim and Tewari (2020), whose algorithm is approximated by transformers through the analysis of Lin et al. (2024). The bound for ours shows the optimal rate derived in Besbes et al. (2014).

optimal online RL algorithms like LinUCB and Thompson Sampling in a multi-armed bandit, as well as UCB value iteration for tabular Markov decision processes.

Despite these encouraging results, an important open challenge remains: *Can transformers adapt to non-stationary environments?* Most existing theoretical analyses assume a fixed task or reward distribution during the transformer's in-context inference (Mao et al., 2021; Domingues et al., 2021). However, in many real-world scenarios, the environment is non-stationary—the reward-generating process evolves over time, violating the assumptions of a static underlying model. In multi-armed bandits, for instance, the true reward probabilities of the arms might drift or change abruptly due to seasonality or shifts in user preferences. Classical algorithms for non-stationary bandits—such as sliding-window UCB, periodic resets, or change-point detection—are known to achieve sublinear *dynamic regret* that closely tracks the moving optimum (Besbes et al., 2014; Garivier and Moulines, 2011; Wei and Luo, 2021; Mao et al., 2021; Cheung et al., 2020). However, the behavior of transformers in non-stationary environments remains largely unexplored.

**Our study.**  In this paper, we investigate whether a transformer, trained in a supervised in-context learning setting, can learn in non-stationary environments. Specifically, we introduce a transformer architecture designed to adapt to non-stationarity, and we analyze both the approximation error and the generalization error arising from learning it from data. Based on these results, we derive an upper bound for the dynamic regret of the transformer and show its optimality.

Our main contributions are summarized as follows:

1. *Regret optimality in non-stationary bandits:* We prove that a transformer acting as an in-context bandit learner can achieve cumulative regret matching the minimax optimal rate for non-stationary environments (e.g., $T^{2/3}$ for bounded changes) (Besbes et al., 2014). Rates are summarized in Table 1.
2. *Generalization analysis under distribution shifts:* Extending the offline-to-online generalization theory of Lin *et al.* (2024) (Lin et al., 2024), we derive conditions on pretraining data diversity and model size needed to guarantee low regret in new non-stationary environments.
3. *Architectural requirements for implementation:* We identify key features—depth, attention heads, and activations—that enable a transformer to forget outdated information and implement restart-style strategies akin to handling non-stationary environments.
4. *Novel proof technique via internal algorithm selection:* We develop a proof approach that treats the transformer's hidden state as maintaining multiple hypothesis policies and uses learned randomness in self-attention to sample among them, yielding tight regret bounds.

## 1.1  Related Works

**In-context learning**  In-context learning (ICL) was first introduced by Brown et al. (2020), allowing large language models (LLMs) to learn tasks by using a few demonstration examples without fine-tuning or updating model parameters. Since its introduction, researchers have studied the properties and mechanisms of ICL in depth (Mao et al., 2024; Zhao et al., 2024; Mosbach et al., 2023; Singh et al., 2024; Bertsch et al., 2025). For example, Mao et al. (2024) adopts a data generation perspective, showing that skill recognition and skill learning abilities in LLMs can transfer between tasks. Zhao et al. (2024) examines the decision boundaries of LLMs in classification tasks and finds that, while in-context examples can improve accuracy, the models often display fragmented decision boundaries, meaning small input changes can lead to different outputs. Moreover, like fine-tuning,

ICL can be unstable and perform poorly on both in-domain and out-of-domain data, although fine-tuning tends to generalize more effectively (Mosbach et al., 2023).

**Approximation capability of transformers** Transformers have been widely applied to tasks such as reinforcement learning, computer vision, graph processing, and speech recognition (Chen et al., 2021; Lin et al., 2024; Liu et al., 2021; Dosovitskiy et al., 2021; Yun et al., 2019; Min et al., 2022; Dong et al., 2018). For instance, Lin et al. (2024) explores using transformers for in-context reinforcement learning, Chebotar et al. (2023) applies them to model Q-functions for multi-task policies, and Zhao et al. (2022) combines vision transformers with reinforcement learning for video captioning. While transformers have shown great potential, their limitations have also been discussed in the literature (Hahn, 2020; Bhattamishra et al., 2020).

**Non-stationary Reinforcement Learning** Non-stationary reinforcement learning (RL) addresses scenarios where rewards and transitions evolve over time, with the degree of difficulty often characterized by the frequency and magnitude of these changes (Auer et al., 2019; Chen et al., 2019; Cheung et al., 2020). To tackle this challenge, Wei and Luo (2021) propose a method that periodically schedules instances with a restart mechanism, while Cheung et al. (2020) advocate leveraging recent data and employing wider confidence intervals to adapt to shifting conditions. Meta-reinforcement learning approaches have also been explored in non-stationary environments (Bing et al., 2022). Other research efforts address non-stationary settings as well, although many rely on prior knowledge, such as the number or extent of environmental changes, to achieve effective adaptation (Mao et al., 2021; Li and Li, 2019; Domingues et al., 2021).

## 1.2 Notation

We use the following notations throughout the paper. $[s, e]$ for $s, e \in \mathbb{N}$ with $s \leqslant e$ denotes the integer interval $\{s, s+1, \ldots, e\}$. $\Delta(\mathcal{X})$ denotes the space of probability distributions over a finite set $\mathcal{X}$. For sets $\mathcal{X}_1, \ldots, \mathcal{X}_n$, we write $\bigotimes_{i=1}^{n} \mathcal{X}_i := \mathcal{X}_1 \times \cdots \times \mathcal{X}_n$ to denote the Cartesian product. $\odot$ denotes element-wise multiplication, and $\bigoplus$ denotes concatenation. $\|\cdot\|_{\mathrm{op}}$ denotes the operator norm of a matrix. For an event $E$, we use $\mathbb{1}_E[x]$ to denote an indicator function on $E$, which equals 1 if $x \in E$, and 0 otherwise. $\mathcal{O}(\cdot)$ is Landau's Big-O notation which hides some absolute constant, and $\widetilde{\mathcal{O}}(\cdot)$ additionally hides logarithmic terms. $[a; b]$ denotes that $a$ and $b$ are stacked vertically.

## 2 Problem Setup

### 2.1 Learning Setup

We first define an environment of reinforcement learning following Lin et al. (2024). Let $T \in \mathbb{N}$ be a number of rounds, and define a tuple of state/action/reward spaces $\{\mathcal{S}_t, \mathcal{A}_t, \mathcal{R}_t\}_{t \in [T]}$, a transition model $\mathbb{T}_t : \mathcal{S}_t \times \mathcal{A}_t \to \Delta(\mathcal{S}_{t+1})$ with $\mathcal{S}_0, \mathcal{A}_0 = \{\varnothing\}$, and the reward function $\mathsf{R}_t : \mathcal{S}_t \times \mathcal{A}_t \to \Delta(\mathcal{R}_t)$. We assume a finite action space $A = |\mathcal{A}_t| < \infty$.

Next, we define a learning scheme for the environment. Let $D_t = (s_j, a_j, r_j)_{j=1}^{t} \subset \bigotimes_{j=1}^{t} \mathcal{S}_j \times \mathcal{A}_j \times \mathcal{R}_j$ be a sequence of observed state/action/reward tuples, where $r_t = \mathsf{R}_t(s_t, a_t)$. We define an algorithm $\mathsf{alg} : (\bigotimes_{j=1}^{t-1} \mathcal{S}_j \times \mathcal{A}_j \times \mathcal{R}_j) \times \mathcal{S}_t \to \Delta(\mathcal{A}_t)$. Then, we obtain a distribution function over a trajectory $D_T$ as

$$\mathbb{P}_{\mathsf{alg}}(D_T) = \prod_{t=1}^{T} \mathbb{T}_{t-1}(s_t|s_{t-1}, a_{t-1})\mathsf{alg}(a_t|D_{t-1}, s_t)\mathsf{R}_t(r_t|s_t, a_t). \tag{1}$$

For training, we assume a base algorithm $\mathsf{alg}_B$ that provides high-quality actions and serves as a guiding policy for training the transformer-based algorithm.

Given $N$ i.i.d. trajectories $D_T^i = (s_1^i, a_1^i, r_1^i, ..., s_T^i, a_T^i, r_T^i)_{i \in [N]} \sim \mathbb{P}_{\mathsf{alg}_0}$ with an offline algorithm $\mathsf{alg}_0$, we augment each $D_T^i$ by $\overline{a}_t^i \sim_{i.i.d.} \mathsf{alg}_B(\cdot|D_{t-1}^i, s_t^i)_{t \in [T]}$ with a base algorithm $\mathsf{alg}_B$ and denote it as $\overline{D}_T^i$. The base algorithm can observe the full trajectory and environment to generate actions for the supervised learning of $\mathsf{alg}_0$.

Using the reward function defined above, we further define non-stationary environments following Wei and Luo (2021).

**Definition 1** (Non-stationary measure). *A function $\Delta : [T] \to \mathbb{R}$ is defined as a non-stationarity measure if it satisfies*

$$\Delta(t) \geqslant \max_{s \in \mathcal{S}, a \in \mathcal{A}} |\mathsf{R}_t(s, a) - \mathsf{R}_{t+1}(s, a)| \tag{2}$$

*for all $t$. Given any interval $\mathcal{I} = [s, e]$, we define $\Delta_{\mathcal{I}} := \sum_{\tau=s}^{e-1} \Delta(\tau)$ and $J_{\mathcal{I}} := 1 + \sum_{\tau=s}^{e-1} \mathbb{1}[\Delta(\tau) \neq 0]$. We denote $\Delta = \Delta_{[1,T]}$ and $J = J_{[1,T]}$ as the amount and number of changes in the environment, respectively.*

Our goal is to develop a bound on the dynamic regret of transformers under the non-stationary setup. In preparation, we define the maximal achievable reward $r_t^* := \max_{s \in \mathcal{S}_t, a \in \mathcal{A}_t} \mathsf{R}_t(s, a)$ at round $t$, its accumulated version $R^*(T) = \sum_{\tau=1}^{T} r_\tau^*$, and the expected reward $R_{\mathsf{alg}}(T) = \mathbb{E}_{D_T \sim \mathbb{P}_{\mathsf{alg}}} \left[ \sum_{\tau=1}^{T} r_\tau \right]$ for an algorithm $\mathsf{alg}$. The regret with an algorithm $\mathsf{alg}$ is then defined as

$$\mathfrak{R}_{\mathsf{alg}}(T) := R^*(T) - R_{\mathsf{alg}}(T). \tag{3}$$

We will show that transformers can effectively solve the non-stationary data problem.

## 2.2 Transformer

We define the architecture of a transformer. Given an input sequence $\mathbf{H} \in \mathbb{R}^{d \times n}$, where $d$ is the feature dimension and $n$ is the sequence length, consider a masked attention layer with $M$ heads. Denote the parameters as $\{(\mathbf{V}_m, \mathbf{Q}_m, \mathbf{K}_m)\}_{m \in [M]} \subset (\mathbb{R}^{d \times d})^{\times 3}$. The output of the masked attention layer is denoted as $\overline{\mathbf{H}} = \mathrm{Attn}_\theta(\mathbf{H}) = [\overline{\mathbf{h}}_1, \dots, \overline{\mathbf{h}}_n] \in \mathbb{R}^{d \times n}$, where each $\overline{\mathbf{h}}_i$ for $i \in [n]$ is given by:

$$\overline{\mathbf{h}}_i = [\mathrm{Attn}_\theta(\mathbf{H})]_i = \mathbf{h}_i + \frac{1}{i} \sum_{m=1}^{M} \sum_{j=1}^{i} \mathrm{ReLU}\left(\langle \mathbf{Q}_m \mathbf{h}_i, \mathbf{K}_m \mathbf{h}_j \rangle\right) \cdot \mathbf{V}_m \mathbf{h}_j \in \mathbb{R}^d,$$

where $\mathrm{ReLU}(x) = \max\{0, x\}$. With slight abuse of notation, we write $\overline{\mathbf{h}}_i = \mathrm{Attn}_\theta(\mathbf{h}_i, \mathbf{H})$. Next, the attention output is fed into a feedforward MLP layer $\mathrm{MLP}_{\theta_{\mathrm{mlp}}}(\cdot)$ with parameter $\theta_{\mathrm{mlp}} = (\mathbf{W}_1, \mathbf{W}_2) \in \mathbb{R}^{d' \times d} \times \mathbb{R}^{d' \times d}$, where $\overline{\overline{\mathbf{h}}}_i = \overline{\mathbf{h}}_i + \mathbf{W}_2 \cdot \mathrm{ReLU}(\mathbf{W}_1 \overline{\mathbf{h}}_i) \in \mathbb{R}^d$, and $D'$ is the hidden dimension of the MLP layer, a transformer with $L$ layers is then defined as $\mathrm{TF}_\theta(\mathbf{H}) = \mathbf{H}^{(L)}$, where for $\ell \in [L]$:

$$\mathbf{H}^{(\ell)} = \mathrm{MLP}_{\theta_{\mathrm{mlp}}^{(\ell)}}\left(\mathrm{Attn}_{\theta_{\mathrm{attn}}^{(\ell)}}(\mathbf{H}^{(\ell-1)})\right) \in \mathbb{R}^{d \times n}.$$

The transformer parameters are collectively denoted as $\boldsymbol{\theta} = (\theta_{\mathrm{attn}}^{(1:L)}, \theta_{\mathrm{mlp}}^{(1:L)})$. The attention parameters are $\theta_{\mathrm{attn}}^{(\ell)} = \{(\mathbf{V}_m^{(\ell)}, \mathbf{Q}_m^{(\ell)}, \mathbf{K}_m^{(\ell)})\}_{m \in [M]} \subset \mathbb{R}^{d \times d}$, and the MLP parameters are $\theta_{\mathrm{mlp}}^{(\ell)} = (\mathbf{W}_1^{(\ell)}, \mathbf{W}_2^{(\ell)}) \in \mathbb{R}^{d' \times d} \times \mathbb{R}^{d \times d'}$. The norm of a transformer $\mathrm{TF}_{\boldsymbol{\theta}}$ is denoted as

$$\|\boldsymbol{\theta}\| := \max_{\ell \in [L]} \left\{ \max_{m \in [M]} \left\{ \|\mathbf{Q}_m^{(\ell)}\|_{\mathrm{op}}, \|\mathbf{K}_m^{(\ell)}\|_{\mathrm{op}} \right\} + \sum_{m=1}^{M} \|\mathbf{V}_m^{(\ell)}\|_{\mathrm{op}} + \|\mathbf{W}_1^{(\ell)}\|_{\mathrm{op}} + \|\mathbf{W}_2^{(\ell)}\|_{\mathrm{op}} \right\}, \tag{4}$$

where $\| \cdot \|_{\mathrm{op}}$ is the operator norm.

## 2.3 Algorithm induced by transformers in In-Context Learning

Let $s_t \in \mathcal{S}_t$ and $(a_t, r_t) \in \mathcal{A}_t \times \mathcal{R}_t$ be embedded by $\mathtt{h}: \bigcup_{t \in [T]} \mathcal{S}_t \cup \bigcup_{t \in [T]} (\mathcal{A}_t \times \mathcal{R}_t) \to \mathbb{R}^d$ such that $\mathtt{h}(s_t) \in \mathbb{R}^d$ and $\mathtt{h}(a_t, r_t) \in \mathbb{R}^d$. For the input $\mathbf{H} = [\mathtt{h}(s_1), \mathtt{h}(a_1, r_1), \dots, \mathtt{h}(a_{t-1}, r_{t-1}), \mathtt{h}(s_t)] \in \mathbb{R}^{d \times (2t-1)}$, the transformer outputs $\overline{\mathbf{H}} = \mathrm{TF}_\theta(\mathbf{H}) = [\overline{\mathbf{h}}_1, \overline{\mathbf{h}}_2, \dots, \overline{\mathbf{h}}_{2t-1}] \in \mathbb{R}^{d \times (2t-1)}$. By introducing a linear mapping $\mathtt{A} \in \mathbb{R}^{A \times d}$ to extract a distribution over $\mathcal{A}_t$ with $|\mathcal{A}_t| = A$ actions, the algorithm induced by the transformer is defined as:

$$\mathsf{alg}_\theta(\cdot | D_{t-1}, s_t) = \mathrm{softmax}\left(\mathtt{A} \cdot \mathrm{TF}_\theta(\mathbf{H})_{-1}\right). \tag{5}$$

For convenience, we denote $\mathsf{alg}_{\boldsymbol{\theta}} = \mathsf{alg}_{\boldsymbol{\theta}}(\cdot | D_{t-1}, s_t)$.

In supervised pretraining, the transformer parameters are learned by maximizing the log-likelihood over the algorithm class $\{\mathsf{alg}_\theta\}_{\theta \in \Theta}$:

$$\widehat{\theta} = \underset{\theta \in \Theta}{\operatorname{argmax}} \frac{1}{N} \sum_{i=1}^{N} \sum_{t=1}^{T} \log \mathsf{alg}_\theta(\bar{a}_t^i | D_{t-1}^i, s_t^i), \tag{6}$$

where $\Theta := \{\boldsymbol{\theta} = (\theta_{\mathrm{attn}}^{(1:L)}, \theta_{\mathrm{mlp}}^{(1:L)})\}$ is the finite parameter class of transformers.

## 3 Main Result

We show that under non-stationary environments, transformers trained with $\widehat{\theta}$ in (6) can approximate the reduced base algorithm $\overline{\mathsf{alg}}_B = \mathbb{E}_{D_T \sim \mathsf{alg}_0}[\mathsf{alg}_B^t(\cdot | D_T, M) | D_{t-1}, s_t]$. This is achieved by bounding the dynamic regret of transformers.

### 3.1 Preparation

We first introduce definitions required for the approximation theorem. In particular, we define the covering number to quantify the complexity of a class of algorithms. This concept is standard in learning theory (Anthony and Bartlett, 2009) and has been used in analyses of transformers (Lin et al., 2024).

**Definition 2** (Covering number). *A finite subset $\Theta_0 \subseteq \Theta$ is called a $\rho$-cover of the algorithm class $\{\mathsf{alg}_\theta : \boldsymbol{\theta} \in \Theta\}$ if, for every $\boldsymbol{\theta} \in \Theta$, there exists $\boldsymbol{\theta}_0 \in \Theta_0$ such that:*

$$\left\| \log \mathsf{alg}_{\boldsymbol{\theta}_0}(\cdot | D_{t-1}, s_t) - \log \mathsf{alg}_{\boldsymbol{\theta}}(\cdot | D_{t-1}, s_t) \right\|_\infty \leqslant \rho, \quad \forall D_{t-1}, s_t, t \in [T].$$

*The covering number $\mathcal{N}_\Theta(\rho)$ is the minimal cardinality of any such $\rho$-cover $\Theta_0$.*

Next, we define the distribution ratio as a measure of discrepancy between two algorithms, which is used in Lin et al. (2024).

**Definition 3** (Distribution ratio). *The distribution ratio of algorithms $\mathsf{alg}_1$ and $\mathsf{alg}_2$ is defined as:*

$$\mathcal{R}_{\mathsf{alg}_1, \mathsf{alg}_2} := \mathbb{E}_{D_T \sim \mathbb{P}_{\mathsf{alg}_1}} \left[ \prod_{t=1}^{T} \frac{\mathsf{alg}_1(a_t | D_{t-1}, s_t)}{\mathsf{alg}_2(a_t | D_{t-1}, s_t)} \right].$$

Many reinforcement learning algorithms generate an auxiliary value at each round, for instance the upper confidence bound (UCB) in UCB-based algorithms. Let $\widetilde{\mathsf{R}}_t(s_t, a_t)$ denote such an auxiliary value at round $t$, under state $s_t \in \mathcal{S}_t$ and action $a_t \in \mathcal{A}_t$. For simplicity, we write $\widetilde{r}_t := \widetilde{\mathsf{R}}_t(s_t, a_t)$. In transformer-based models, this auxiliary value can be interpreted as the output probability distribution that guides action selection. To introduce the approximation theorem, we introduce the following assumptions used in the non-stationary analysis, e.g., Wei and Luo (2021).

**Assumption 1** (Non-stationarity). *For all $\boldsymbol{\theta} \in \Theta$, $\mathsf{alg}_\theta$ outputs an auxiliary quantity $\widetilde{r}_t \in [0, 1]$ at the beginning of each round $t$. There exists a non-stationarity measure $\Delta$ and a non-increasing function $\rho : [T] \to \mathbb{R}$ such that running $\mathsf{alg}_\theta$ satisfies that without knowing $\Delta_{[1,t]}$, $\mathsf{alg}_\theta$ ensures with probability at least $1 - 1/T$:*

$$\widetilde{r}_t \geqslant \min_{\tau \in [1,t]} r_\tau^* - \Delta_{[1,t]} \quad \text{and} \quad \frac{1}{t} \sum_{\tau=1}^{t} (\widetilde{r}_\tau - r_\tau) \leqslant \rho(t) + \Delta_{[1,t]}$$

*for all $t \in [T]$, provided that $\Delta_{[1,t]} \leqslant \rho(t)$ and $\rho(t) \geqslant 1/\sqrt{t}$. Additionally, we assume the function $C(t) := t\rho(t)$ is non-decreasing.*

In addition, we assume that the base algorithm generating the observation sequence can be approximately realized by a transformer, which is the following assumption. This assumption has been proved under several traditional RL algorithms such as LinUCB and Thompson Sampling (Lin et al., 2024). For additional discussion of this assumption, see Appendix A.

**Assumption 2** (Realizability). *Given a reinforcement learning algorithm $\mathsf{alg}_B$, for any $\varepsilon > 0$, there exist a parameter $\boldsymbol{\theta}$ such that the following holds: there exist constants $D_0, M_0, L_0, D_0', C_0 > 0$, which depend on $\varepsilon$, such that a transformer $\mathsf{TF}_{\boldsymbol{\theta}}$ with $D = D_0$, $M = M_0$, $L = L_0$, $D' = D_0'$, $\|\|\boldsymbol{\theta}\|\| = C_0$, satisfies*

$$\left| \log \mathsf{alg}_{\boldsymbol{\theta}}(a_{t,k}|D_{t-1}, s_t) - \log \overline{\mathsf{alg}}_B(a_{t,k}|D_{t-1}, s_t) \right| < \varepsilon \tag{7}$$

*for all rounds $t$ and actions $a_{t,k}$.*

## 3.2 Regret Bound

As our main result, we develop an upper bound on the dynamic regret of the transformer-induced policy $\mathsf{alg}_{\widehat{\boldsymbol{\theta}}}$. This theorem quantifies how well a transformer-based policy performs in a non-stationary environment compared to an optimal baseline algorithm. The proof is provided in Appendix D.2.

**Theorem 1.** *Let $\mathsf{alg}_B$ be a reinforcement learning algorithm satisfying Assumptions 1 and 2. Suppose Assumption 1 holds with $C(t) = t^p$ for some $p \in [1/2, 1)$, and let $\widehat{\theta}$ be a solution of (6). Assume that $|r_t| \leqslant 1$ holds almost surely, and define $\mathcal{N}_\Theta := \mathcal{N}_\Theta((NT)^{-2})$. Then, for any $\varepsilon > 0$, the dynamic regret of $\mathsf{alg}_{\widehat{\theta}}$ induced by a transformer with $D \leqslant D_0 + 10$, $M = \max\{M_0, 2\}$, $L = L_0$, $D' = \mathcal{O}(\max\{D_0', T \log_2 T/\varepsilon\})$, $\|\|\widehat{\boldsymbol{\theta}}\|\| = \widetilde{\mathcal{O}}(C_0 + \sqrt{T \log_2 T/\varepsilon})$ satisfies the following with probability at least $1 - \max\{\delta, 1/T\}$ with $\mathcal{R} := \mathcal{R}_{\mathsf{alg}_{\widehat{\theta}}, \overline{\mathsf{alg}}_B}$:*

$$\mathfrak{R}_{\mathsf{alg}_{\widehat{\theta}}}(T) = \widetilde{\mathcal{O}}\Bigg( \underbrace{T^2\sqrt{\mathcal{R}}\left( \sqrt{\frac{\log(\mathcal{N}_\Theta T/\delta)}{N}} + \sqrt{\varepsilon} \right)}_{=:T_{\mathrm{comp}}} + \underbrace{\varepsilon A T}_{=:T_{\mathrm{aprx}}} + \underbrace{\min\left\{ \sqrt{JT}, \Delta^{1/3}T^{2/3} + \sqrt{T} \right\}}_{=:T_{\mathrm{algo}}} \Bigg). \tag{8}$$

This bound consists of three main components $T_{\mathrm{comp}}$, $T_{\mathrm{aprx}}$, and $T_{\mathrm{algo}}$. $T_{\mathrm{comp}}$ captures the effect of pretraining the transformer on $N$ supervised samples to learn the base distribution. $T_{\mathrm{aprx}}$ represents the approximation error incurred when using the transformer to imitate the base policy in the non-stationary setting. $T_{\mathrm{algo}}$ corresponds to the intrinsic regret of the base algorithm in a non-stationary environment.

Theorem 1 shows that the transformer-based algorithm performs nearly optimally even in non-stationary environments, with the error diminishing as the training dataset grows and the environment stabilizes. Specifically, when the number of observations $N$ is sufficiently large and the approximation error $\varepsilon$ is sufficiently small, the regret bound in (8) is dominated by the third term $T_{\mathrm{algo}}$. To further clarify this, we present the following corollary, whose proof is contained in Appendix D.3.

**Corollary 2.** *Consider the setup in Theorem 1. There exists a universal constant $C$ such that, for $N \geqslant CT^3 \log T$, and $\varepsilon \leqslant T^{-3}$, the dynamic regret of $\mathsf{alg}_{\widehat{\theta}}$ satisfies*

$$\mathfrak{R}_{\mathsf{alg}_{\widehat{\theta}}}(T) = \widetilde{\mathcal{O}}\left( \min\left\{ \sqrt{JT}, \Delta^{1/3}T^{2/3} + \sqrt{T} \right\} \right). \tag{9}$$

This result implies that, with a sufficiently large dataset and a sufficiently expressive transformer (small $\varepsilon$), the transformer-based policy achieves the optimal dynamic regret in non-stationary environments, matching the bounds established by prior works (Wei and Luo, 2021). Intuitively, smaller $\varepsilon$ corresponds to larger transformer architectures, as indicated in Theorem 1. Hence, with enough data and a suitably large architecture, the transformer is capable of realizing the optimal strategy that minimizes dynamic regret.

While the above bounds could also be achieved with a standard transformer, for the sake of simpler proofs we use a *non-continuous transformer*. Structurally, this model is equivalent to a regular transformer except that its inputs are duplicated a few times to form an augmented input sequence. The motivation for this construction, along with a discussion on how the proofs extend to standard transformers, is provided in Appendix E.2. An abstract illustration of the non-continuous transformer is shown in Figure 1.

# 4 Proof Outline

In this section, we provide an overview of the proof of Theorem 1. The most critical component is the analysis of the approximation error $T_{\text{aprx}}$. To this end, we first introduce common operations and algorithmic structures used to handle non-stationary data, and then show how a transformer can approximate these operations effectively.

## 4.1 Operation for Non-Stationary Environments

In non-stationary environments, maintaining strong performance requires **limiting the reliance on historical data**. Existing approaches primarily employ two strategies: using a *window scheduler*, where only data within a sliding window is available to the model (Cheung et al., 2022; Trovo et al., 2020), and a *test-and-restart mechanism*, which resets the algorithm when significant changes in the environment are detected (Gomes et al., 1998; Cheung et al., 2020; Cayci et al., 2020; Wei and Luo, 2021; Mao et al., 2021). Among these approaches, the *MASTER* algorithm (Wei and Luo, 2021) stands out. MASTER not only achieves near-optimal performance in non-stationary environments but also does not require prior knowledge of the environment's change parameters, such as the number or magnitude of changes.

We briefly describe the window scheduler and the restarting operation in MASTER, while their formal definitions will be provided in Appendix E.1. The overview of the proof is illustrated in Figure 1.

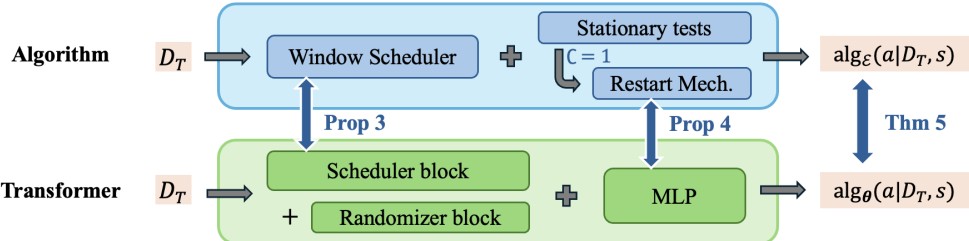

Figure 1: Overview of the proof structure for the approximation analysis (Theorem 5). The blue box represents the algorithmic operations for handling non-stationarity, while the green box represents the transformer approximating the algorithm. Each block or sub-architecture of the transformer corresponds to a specific operation of the algorithm.

**Window scheduler:** Denote $n + 1$ copies of trajectories $D_T$ as $\{D_T^{(i)}\}_{i=0}^n$. Given window sizes $\{W^{(i)}\}_{i=0}^n$ and probabilities $\{p^{(i)}\}_{i=0}^n$, we define the following two modules: (i) a masking operator, and (ii) a selection operator.

First, the *masking operator* $\sigma_1$ maps the input trajectory $D_T$ to a randomly masked version of the copies $\{D_T^{(i)}\}_{i=0}^n$, using the window sizes and probabilities. Formally,

$$\sigma_1(D_T) = \bigoplus_{i=0}^n \left( \mathbf{m}^{(i)} \odot D_T^{(i)} \right),$$

where $\mathbf{m}^{(i)} \in \{0, 1\}^T$ is a binary mask for copy $i$ defined elementwise as

$$m_k^{(i)} = \begin{cases} \text{Bernoulli}(p^{(i)}) & \text{if } \exists\, t \equiv 1 \pmod{2^i},\ k \in [t, t + W^{(i)} - 1] \\ 0 & \text{otherwise} \end{cases}.$$

This operator probabilistically selects candidate time points of interest within each copy based on the window size $W^{(i)}$ and time position $t$.

Second, the *selection operator* $\sigma_2$ chooses a single active trajectory instance among the masked copies:

$$\sigma_2 \left( \{(s_t^{(i)}, a_t^{(i)}, r_t^{(i)})\}_{i=0}^n \right) = \left[ \mathbf{0}; \ldots; \mathbf{0}; (s_t^{(k)}, a_t^{(k)}, r_t^{(k)}); \mathbf{0}; \ldots; \mathbf{0} \right],$$

where $k = \min\{j \in \{0, \dots, n\} : (s_t^{(j)}, a_t^{(j)}, r_t^{(j)}) \neq \mathbf{0}\}$ is the index of the first non-zero entry (order-$k$). The selected instance is called "active", while the others are "inactive". This ensures that only one instance is active at each round, and it is always the lowest-order scheduled instance.

Finally, the *window scheduler* (WS) is defined as:
$$\text{WS}(D_T) = \sigma_2 \circ \sigma_1(D_T).$$
The window scheduler stochastically schedules multi-scale windows to restrict the algorithm to recent data (Figure 2).

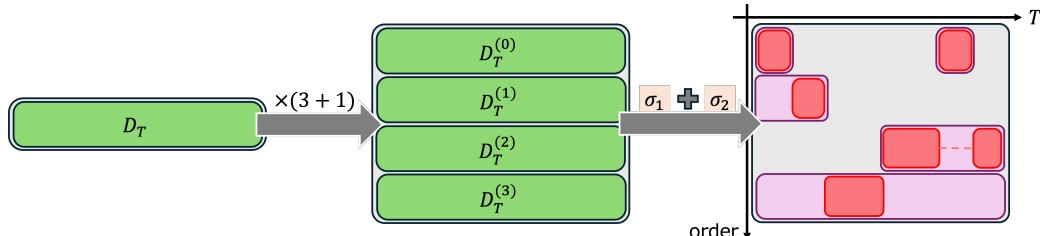

Figure 2: Illustration of WS when $n = 3$. Purplish blocks represent instances scheduled by $\sigma_1$, while reddish blocks represent the active instances selected by $\sigma_2$. Reddish blocks connected by a dashed line are concatenated.

**Stationary Test and Restart Mechanism:** After passing through the window scheduler WS, the base algorithm interacts with the environment, receives a reward, and performs a *stationary test*, which outputs a change-detection signal $\mathsf{C}_t \in \{0, 1\}$. If a significant change in the environment is detected ($\mathsf{C}_t = 1$), the algorithm is restarted. We denote the output of WS as the sequence of active instances $\{(s_t', a_t', r_t')\}_{t=1}^{T}$. We then define the auxiliary memory as $\mathcal{A}_t := \{(s_p', a_p', r_p', \widetilde{r}_p)\}_{p=1}^{t}$, where the auxiliary value $\widetilde{r}_t$ is added at each step.

Given the change-detection signal $\mathsf{C}_t \in \{0, 1\}$ and auxiliary memory $\mathcal{A}_t$, the restart mechanism (RM) is defined as:
$$\text{RM}(\mathcal{A}_{t+1}) = \begin{cases} \mathcal{A}_0 & \text{if } \mathsf{C}_t = 1 \\ \mathcal{U}_\mathcal{A}(\mathcal{A}_t, D_t) & \text{otherwise} \end{cases}$$
where $\mathcal{A}_0$ is an empty/reset auxiliary state (all counts=0, rewards=$\varnothing$), and $\mathcal{U}_\mathcal{A}$ updates the auxiliary memory (e.g., appending new rewards). Typically, $\widetilde{r}_t$ is evaluated to determine when a restart is necessary.

It is worth noting that the sliding window and restart mechanisms used in previous works (e.g., (Cheung et al., 2022; Trovo et al., 2020; Cayci et al., 2020)) can be considered special cases of WS and RM. We discuss their equivalence in Appendix E.4. As a result, this approximation can be extended to a wide range of algorithms designed to handle non-stationarity.

### 4.2 Transformers for Approximating the Operations

Our goal is to demonstrate that transformers can approximate the above techniques and the overall algorithm. Given a base algorithm $\text{alg}_B$, we have the following statement.

**Proposition 3** (Approximating WS). *For any small $\varepsilon > 0$, there exists a transformer $\text{TF}_{\boldsymbol{\theta}}(\cdot)$ with $D = \mathcal{O}(1), L = \mathcal{O}(1), M = \mathcal{O}(1/\sqrt{\varepsilon}), D' = \mathcal{O}(\sqrt{T \log_2 T/\varepsilon})$ such that the algorithm $\text{alg}_{\boldsymbol{\theta}}$, defined in (5), can generate a random number drawn from a uniform distribution and satisfies*
$$|\log \text{alg}_{\boldsymbol{\theta}}(a_{t,k}|D_{t-1}, s_t) - \log \text{alg}_B(a_{t,k}|\text{WS}(D_{t-1}), s_t)| < \varepsilon, \quad \forall t \in [T], k \in [A]. \tag{10}$$

This proposition shows the existence of a transformer block (called the scheduler block) that approximates the window scheduler WS. In the construction, we also include a block that extracts randomness from the data to implement the stochastic scheduling in WS. The proof of Proposition 3 is provided in Appendix E.3.

Furthermore, since RM involves only zeroing out certain values and applying a mask $\mathbb{1}_{>0}[\cdot]$ to detect changes, its approximation by a transformer follows directly. Details for approximating RM are included in Appendix E.1 as part of the overall proof.

**Proposition 4** (Approximating RM). *The restart mechanism* RM *can be implemented by a two-layer MLP with* $D' = \mathcal{O}(1)$.

Finally, we consider an expert algorithm $\text{alg}_E$, which incorporates both the window scheduler and the restart mechanism to handle the non-stationary environment. The full details of this algorithm are provided in Section C. Then, the following theorem holds:

**Theorem 5.** *For any* $\varepsilon > 0$, *there exists a transformer* $\text{TF}_{\boldsymbol{\theta}}$ *with* $D \leqslant D_0 + 10, M = \max\{M_0, 2\}, L = L_0, D' = \mathcal{O}(\max\{D_0', T\log_2 T/\varepsilon\}), \|\|\boldsymbol{\theta}\|\| = \widetilde{\mathcal{O}}(C_0 + \sqrt{T\log_2 T/\varepsilon})$, *such that the transformer-based algorithm* $\text{alg}_{\boldsymbol{\theta}}$ *defined in* (5) *satisfies*

$$\left|\log \text{alg}_{\boldsymbol{\theta}}(a_{t,k}|D_{t-1}, s_t) - \log \overline{\text{alg}}_E(a_{t,k}|D_{t-1}, s_t)\right| < \varepsilon, \quad \forall t \in [T], k \in [A]. \tag{11}$$

With this result, we obtain the approximation error $T_{\text{aprx}}$. The entire proof is shown in Figure 1. The proofs of Theorem 5 is contained in Appendix D.4.

Together with the analyses of the training error $T_{\text{comp}}$ and the algorithmic regret $T_{\text{algo}}$, Theorem 5 shows that transformers can effectively replicate the techniques required for handling non-stationary environments, completing the argument for Theorem 1. Full technical details are provided in the supplementary material.

## 5  Experiment

In this section, we evaluate transformers and other algorithms in a linear bandit setting. The stochastic linear bandit framework is given by $M = (w^*, \mathcal{E}, \mathcal{A}_1, \ldots, \mathcal{A}_T)$. At each round $t \in [T]$, the learner selects an action $a_t \in \mathbb{R}^d$ from the set $\mathcal{A}_t = \{a_{t,1}, \ldots, a_{t,A}\}$, which may vary over time. The learner then receives a reward $r_t = \langle a_t, w^* \rangle + \varepsilon_t$, where $\varepsilon_t \sim_{i.i.d.} \mathcal{E}$ and $w^* \in \mathbb{R}^d$ is unknown. The problem generalizes by setting $s_t = \mathcal{A}_t$, with the state transitioning deterministically to $s_{t+1}$ regardless of the action.

We compare transformers against Linear UCB (LinUCB) and Thompson Sampling (TS), as well as MASTER (Wei and Luo, 2021) combined with LinUCB/TS (denoted as expert algorithms) under environments with varying degrees of non-stationarity. In our experiments, we set $d = 32, A = 10$, $\varepsilon_t \sim \mathcal{N}(0, 1.5^2)$, and $w^* \sim \text{Unif}([0,1]^d)$. We consider two types of environments:

(1) Low Non-Stationarity: Models are evaluated over 1,000 rounds, with elevated rewards in $t \in [50, 100] \cup [350, 400]$ scaled to $r_t \in [3, 4]$, and the remaining rewards in $r_t \in [0, 1]$. Training data consists of 100,000 samples with normalized rewards $r_t \in [0, 1]$.

(2) High Non-Stationarity: The reward is defined as $r_t = (\langle a_t, w^* \rangle + \varepsilon_t)\cos(2\pi bt)$. For training, we generate 100,000 samples for each $b \in \{0.005, 0.01, 0.015, 0.02\}$. For evaluation, we test on unseen environments with $b \in \{0.018, 0.025\}$, running 200 rounds per environment.

For transformer models, we use GPT-2 with $L = 16$ layers and $M = 16$ attention heads, trained for 200 epochs. The objective is to assess generalization to non-stationary environments unseen during training.

From Figure 3, we observe that transformers achieve performance comparable to, and sometimes surpassing, both the expert algorithms and their MASTER variants, attaining near-optimal cumulative regret. Additional experimental results can be found in Appendix B.

## 6  Conclusion

In this study, we have shown that transformers can effectively handle non-stationary environments, achieving near-optimal performance by minimizing dynamic regret. By demonstrating that transformers can implement strategies commonly used for adapting to non-stationarity, we provide a theoretical guarantee for their dynamic regret bound. Our experiments further show that transformers can match, and in some cases outperform, existing expert algorithms. As a limitation, we evaluated transformers empirically only against two RL algorithms and their variants in the linear bandit setting. Future work should explore additional experimental setups and broader algorithm comparisons to further validate our findings.

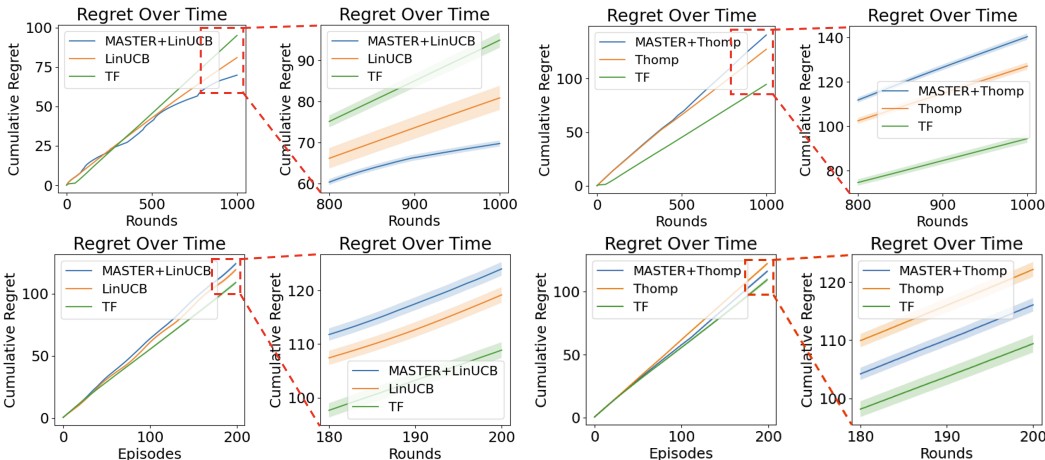

Figure 3: Cumulative regret comparison for LinUCB, Thompson Sampling (TS), MASTER+LinUCB/TS, and transformer (TF) in linear bandits $d = 32$, $A = 10$. The first row corresponds to Low Non-Stationarity environments, while the second row shows High Non-Stationarity environments. Shading indicates the standard deviation of the regret estimates.

## Acknowledgements

Shinji Ito was supported by JSPS KAKENHI (JP25K03184). Masaaki Imaizumi was supported by JSPS KAKENHI (JP24K02904), JST CREST (JPMJCR21D2), and JST FOREST (JPMJFR216I).

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

# A    Limitation and Discussion

**Regret bound**    In Corollary 2, the regret bound of the learned algorithm $\text{alg}_{\widehat{\theta}}$ is guaranteed when the approximation error $\varepsilon$ is sufficiently small—which requires a sufficiently large model size—and when the dataset size $N$ is sufficiently large. Although these requirements may seem inefficient compared to the model sizes and training datasets of traditional RL algorithms, modern LLMs are typically trained with scales that readily meet both conditions.

**Proof method**    To establish the regret bound, we employ a non-continuous transformer, which is structurally equivalent to a standard transformer. This choice is made primarily for proof simplicity. While we also provide a brief description of how the standard transformer can be used for the proof (Appendix E.2), developing a more elegant proof directly in that setting is left for future work.

**Pretraining of transformer**    The regret bound holds without assumptions on the non-stationarity of the training data. A natural direction for future work is to examine how training data with different types and degrees of non-stationarity influence the generality and performance of transformers.

**Assumption 1 and 2**    Our theoretical guarantees rest on Assumptions 1 and 2. The former has been shown to hold for a wide range of RL algorithms (Wei and Luo, 2021), whereas the latter may appear somewhat strong. We emphasize, however, that Assumption 2 builds on prior work demonstrating that transformers can emulate standard reinforcement learning algorithms, such as LinUCB and Thompson Sampling, through in-context learning (Lin et al., 2024). In addition, Furuya et al. (2025) shows that transformers are universal approximators over distributions, which further supports the plausibility of this assumption. We also emphasize that the assumption does not require the transformer to exactly replicate the expert policy. Instead, it assumes that the transformer can learn an amortized algorithm that approximates the expert's output distribution, which is a more relaxed and often more realistic requirement in practice.

# B    Further Experiment Results

In our experiments, we set the confidence scaling parameter $\alpha$ of LinUCB to 1, and the noise variance for Thompson Sampling (TS) to 0.3.

The suboptimality of the models in Low Non-stationary environments and in High Non-stationary environments with $b = 0.018$ (Figure 3) is shown in Figures 4 and 5. We observe that the transformer not only outperforms the expert algorithms but also maintains a consistently low suboptimality rate across rounds.

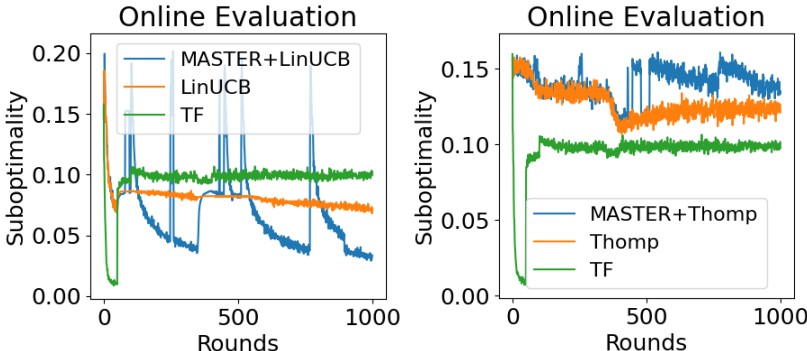

Figure 4:    Suboptimality Comparisons of LinUCB, Thompson Sampling (TS), MASTER+LinUCB/TS, and transformer (TF) in Low Non-stationary environments. Linear bandit with $d = 32$, $A = 10$. Shading indicates the standard deviation of the regret estimates.

Figures 6 and 7 present the results for High Non-stationary environments with $b = 0.025$. We observe that, when the level of non-stationarity becomes too high, the transformer no longer outperforms the other models. This indicates that the model generalizes well to $b = 0.018$, but struggles

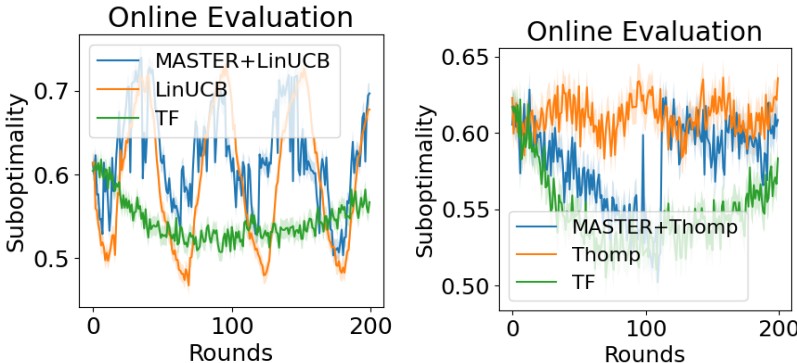

Figure 5: Suboptimality Comparisons of LinUCB, Thompson Sampling (TS), MASTER+LinUCB/TS, and transformer (TF) in High Non-stationary environments ($b = 0.018$). Linear bandit with $d = 32$, $A = 10$. Shading indicates the standard deviation of the regret estimates.

with $b = 0.025$, likely due to limited coverage of highly non-stationary scenarios in the training data.

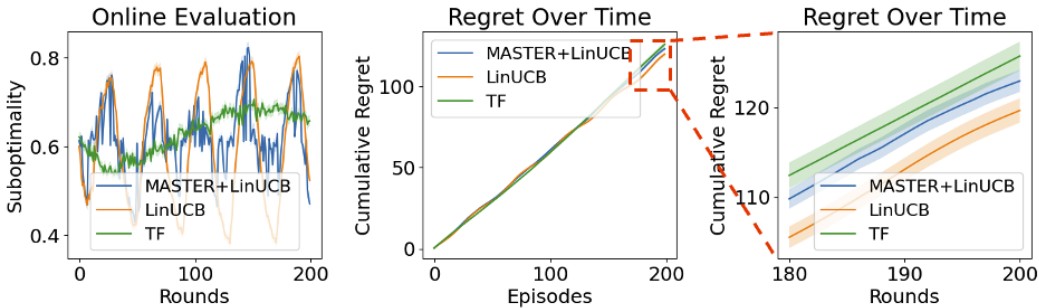

Figure 6: Suboptimality and Cumulative Regret Comparisons of LinUCB, MASTER+LinUCB, and transformer (TF) in High Non-stationary environments ($b = 0.025$). Linear bandit with $d = 32$, $A = 10$. Shading indicates the standard deviation of the regret estimates.

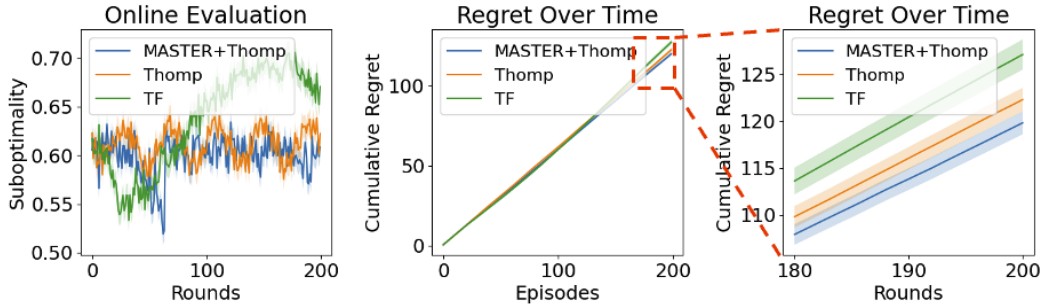

Figure 7: Suboptimality and Cumulative Regret Comparisons of Thompson sampling (TS), MASTER+TS, and transformer (TF) in High Non-stationary environments ($b = 0.025$). Linear bandit with $d = 32$, $A = 10$. Shading indicates the standard deviation of the regret estimates.

## C  MASTER Algorithm

We begin by introducing MALG (Multi-scale ALGorithm) (Wei and Luo, 2021), which extends a base reinforcement learning algorithm, ALG, by running multiple instances at different time scales.

Given an integer $n$ and a non-increasing function $\rho : [T] \to \mathbb{R}$, MALG operates over a time horizon of length $2^n$.

At each time step $\tau$, given an integer $m$, if $\tau$ is a multiple of $2^m$, MALG schedules a new instance of length $2^m$ with probability $\rho(2^n)/\rho(2^m)$. These are referred to as order-$m$ instances. The start and end times of each instance are denoted by $alg.s$ and $alg.e$, respectively. Among all scheduled instances, only the one with the lowest order remains active at any given time, producing an auxiliary value $\widetilde{r}_t$.

After each step, MALG updates the active instance based on observed rewards $R_t$ from the environment. In this framework, all reinforcement learning algorithms generate a specific auxiliary value; for instance, in the LinUCB setting, $\widetilde{r}_t$ corresponds to a score that combines a reward estimate with an uncertainty term.

---

**Algorithm 1:** MALG (Multi-scale ALG)(Wei and Luo, 2021)

**Input:** $n, \rho(\cdot)$
1 **for** $\tau = 0, \ldots, 2^n - 1$ **do**
2     **for** $m = n, n-1, \ldots, 0$ **do**
3         **if** $\tau$ *is a multiple of* $2^m$ **then**
4              With probability $\rho(2^n)/\rho(2^m)$, schedule a new instance *alg* of ALG at scales $2^m$;
5     Run the active instance *alg* to output $\widetilde{r}_\tau$, select an action, and update with feedback.

---

**Algorithm 2:** MALG with Stationarity TEsts and Restarts (MASTER)(Wei and Luo, 2021)

**Input:** $\widehat{\rho}(\cdot)$ where $\widehat{\rho}(t) = 6(\log_2 T + 1)\log(T)\rho(t)$ ($T$: block length)
1 **Initialize** $t \leftarrow 1$
2 **for** $n = 0, 1, \ldots$ **do**
3     Set $t_n \leftarrow t$ and initialize an MALG (Algorithm 2) for the block $[t_n, t_n + 2^n - 1]$;
4     **while** $t < t_n + 2^n$ **do**
5         Run MALG to obtain prediction $\widetilde{r}_t$, select action $a_t$, and receive reward $R_t$;
6         Update MALG with feedback, and set $U_t = \min_{\tau \in [t_n, t]} \widetilde{r}_\tau$;
7         Perform Test 1 and Test 2 (see below);
8         Increment $t \leftarrow t + 1$;
9         **if** *either test returns fail* **then**
10             restart from Line 2;

11 **Test 1**: If $t = alg.e$ for some order-$m$ alg and $\frac{1}{2^m}\sum_{\tau=alg.s}^{alg.e} R_\tau \geqslant U_t + 9\widehat{\rho}(2^m)$, return fail.
12 **Test 2**: If $\frac{1}{t-t_n+1}\sum_{t_n}^{t}(\widetilde{r}_\tau - r_\tau) \geqslant 3\widehat{\rho}(t - t_n + 1)$, return fail.

---

MASTER (MALG with Stationarity Tests and Restarts) further enhances this process by incorporating repeated checks for stability and performance. It uses two key tests: the first ensures that the average reward within any completed instance does not significantly exceed a threshold determined by the auxiliary value $\widetilde{r}_t$; the second verifies that the difference between the auxiliary output $\widetilde{r}_t$ and observed rewards $r_t$ remains bounded on average. If either test fails, MASTER resets the algorithm, thereby maintaining robustness under changing environmental conditions.

## D   Proofs in Section 3

### D.1   Intermediate Statement

We provide the following intermediate statement to show Theorem 1.

**Theorem 6.** *Let Assumption 1 and 2 hold. Then for any small $\varepsilon > 0$, there exists an algorithm* $\text{alg}_{\widehat{\boldsymbol{\theta}}}$ *introduced by a transformer with*

$$D \leqslant D_0 + 10, \quad M = \max\{M_0, 2\}, \quad L = L_0,$$
$$D' = \mathcal{O}(\max\{D_0', T\log_2 T/\varepsilon\}), \quad \|\!|\widehat{\boldsymbol{\theta}}|\!\| = \widetilde{\mathcal{O}}(C_0 + \sqrt{T\log_2 T/\varepsilon}) \tag{12}$$

*satisfying*

$$|R_{\mathsf{alg}_{\hat{\theta}}}(T) - R_{\overline{\mathsf{alg}}_E}(T)| = \tilde{\mathcal{O}}\left(T^2\sqrt{\mathcal{R}_{\mathsf{alg}_{\hat{\theta}},\overline{\mathsf{alg}}_B}}\left(\sqrt{\frac{\log(\mathcal{N}_\Theta T/\delta)}{n}}\right) + \varepsilon AT\right). \tag{13}$$

## D.2 Proof of Theorem 1

As established in Wei and Luo (2021), for any reinforcement learning algorithm $\mathsf{alg}_E$, the combination of MASTER and alg results in a stabilized version: $\overline{\mathsf{alg}}_E$. The dynamic regret of $\mathsf{alg}_{\hat{\theta}}$ can then be expressed as

$$\mathfrak{R}_{\mathsf{alg}_{\hat{\theta}}}(T) \leqslant \underbrace{\left|\mathfrak{R}_{\mathsf{alg}_{\hat{\theta}}}(T) - \mathfrak{R}_{\overline{\mathsf{alg}}_E}(T)\right|}_{=:\mathfrak{A}} + \underbrace{\left|\mathfrak{R}_{\overline{\mathsf{alg}}_E}(T)\right|}_{=:\mathfrak{B}}. \tag{14}$$

$\mathfrak{A}$ can be obtained from Theorem 6, and $\mathfrak{B}$ is bounded as follows:

$$\mathfrak{B} = \tilde{\mathcal{O}}\left(\min\left\{\left(c_1 + \frac{c_2}{c_1}\right)\sqrt{JT}, \ \left(c_1^{2/3} + c_2 c_1^{-4/3}\right)\Delta^{1/3}T^{2/3} + \left(c_1 + \frac{c_2}{c_1}\right)\sqrt{T}\right\}\right),$$

where $C(t) = c_1 t^{1/2} + c_2$ with $c_1, c_2 > 0$ (Wei and Luo, 2021). Setting $c_1 = 1$ and $c_2 \ll 1$, we obtain:

$$\mathfrak{B} = \tilde{\mathcal{O}}\left(\min\left\{\sqrt{JT}, \ \Delta^{1/3}T^{2/3} + \sqrt{T}\right\}\right).$$

$\square$

## D.3 Proof of Corollary 2

It suffices to prove that

$$T^2\sqrt{\mathcal{R}_{\mathsf{alg}_{\hat{\theta}},\overline{\mathsf{alg}}_B}}\left(\sqrt{\frac{\log(\mathcal{N}_\Theta T/\delta)}{n}} + \sqrt{\varepsilon}\right) + \varepsilon AT = \tilde{\mathcal{O}}(\sqrt{T}).$$

According to Theorem 5, we have

$$\frac{\mathsf{alg}_{\hat{\theta}}(a_t|D_{t-1}, s_t)}{\overline{\mathsf{alg}}_B(a_t|D_{t-1}, s_t)} \leqslant e^\varepsilon.$$

Thus, the cumulative distribution ratio over $T$ rounds satisfies

$$\mathcal{R}_{\mathsf{alg}_{\hat{\theta}},\overline{\mathsf{alg}}_B} = e^{\varepsilon T}.$$

Since we assume $\varepsilon \leqslant T^{-3}$, it follows that

$$\mathcal{R}_{\mathsf{alg}_{\hat{\theta}},\overline{\mathsf{alg}}_B} = \mathcal{O}(1).$$

Consequently, we have

$$T^2\sqrt{\mathcal{R}_{\mathsf{alg}_{\hat{\theta}},\overline{\mathsf{alg}}_B}} \cdot \sqrt{\varepsilon} + \varepsilon AT = \mathcal{O}(\sqrt{T}).$$

Next, following Vaart and Wellner (2023), there exist universal constants $C_1$ and $C_2$ such that

$$\mathcal{N}_\Theta = \mathcal{N}_\Theta((NT)^{-2}) \leqslant (2N^2T^2)^{C_1 V(C_2(NT)^{-2}, \Theta)}$$

where $V(\varepsilon, \Theta)$ denotes the fat-shattering dimension of $\Theta$. Let $V(\Theta)$ be the VC-dimension of $\Theta$; then, for any $\varepsilon > 0$, it holds that $V(\varepsilon, \Theta) \leqslant V(\Theta)$ (Vaart and Wellner, 2023). Since $\Theta$ is a finite parameter class, there exists a constant $C_3$ such that $V(\varepsilon, \Theta) \leqslant V(\Theta) \leqslant C_3$ for all $\varepsilon > 0$. Therefore, under the assumption that $N \geqslant CT^3 \log T$, we obtain:

$$\begin{aligned}
T^2\sqrt{\mathcal{R}_{\mathsf{alg}_{\hat{\theta}},\overline{\mathsf{alg}}_B}}\sqrt{\frac{\log(\mathcal{N}_\Theta T/\delta)}{N}} &\leqslant T^2\sqrt{\frac{\log(\mathcal{N}_\Theta T/\delta)}{N}} \\
&\leqslant T^2\sqrt{\frac{(1 + 2C_1C_3)\log T + 2C_1C_3\log N + \log(2^{C_1C_3}/\delta)}{N}} \\
&\leqslant T^2\sqrt{\frac{\tilde{\mathcal{O}}(\log T)}{\mathcal{O}(T^3 \log T)}} \\
&= \tilde{\mathcal{O}}(\sqrt{T})
\end{aligned}$$

which completes the proof.

$\square$

### D.4 Proof of Theorem 5

As shown in Section E, a noncontinuous transformer (Definition 7) with

$$D = ((d+5)(n+1)+5), \quad M = 2, \quad L > 0, \quad D' = T \log_2 T/\varepsilon, \quad \|\boldsymbol{\theta}\| = \tilde{\mathcal{O}}\left(\sqrt{T \log_2 T/\varepsilon}\right),$$

can approximate the MASTER algorithm. Treating any reinforcement learning algorithm that uses MASTER as the expert algorithm $\text{alg}_E$ in Theorem 5, the result follows directly from Assumption 2. $\qquad\square$

### D.5 Proof of Theorem 6

By Theorem 5, we have

$$\log \frac{\text{alg}_{\hat{\boldsymbol{\theta}}}(a_{t,k}|D_{t-1}, s_t)}{\overline{\text{alg}}_E(a_{t,k}|D_{t-1}, s_t)} < \varepsilon$$
$$\Rightarrow \text{alg}_{\hat{\boldsymbol{\theta}}}(a_{t,k}|D_{t-1}, s_t) < e^\varepsilon \cdot \overline{\text{alg}}_E(a_{t,k}|D_{t-1}, s_t)$$
$$\Rightarrow \left|\text{alg}_{\hat{\boldsymbol{\theta}}}(a_{t,k}|D_{t-1}, s_t) - \overline{\text{alg}}_E(a_{t,k}|D_{t-1}, s_t)\right| < (e^\varepsilon - 1),$$

where the last inequality uses the fact that $\text{alg}(\cdot) \leqslant 1$. With a slight abuse of notation, let $r_{t,k}$ denote the reward of the $k$-th action at round $t$. Since $|r_{t,k}| \leqslant 1$ almost surely for all $t \in [T]$, we obtain the policy imitation error between transformers and MASTER as follows:

$$\text{Policy Imitation Error} = \left|\sum_{t=1}^{T} \sum_{k=1}^{A} \left(\text{alg}_{\hat{\boldsymbol{\theta}}}(a_{t,k}|D_{t-1}, s_t) - \overline{\text{alg}}_E(a_{t,k}|D_{t-1}, s_t)\right) r_{t,k}\right|$$
$$\leqslant \sum_{t=1}^{T} \sum_{k=1}^{A} \left|\text{alg}_{\hat{\boldsymbol{\theta}}}(a_{t,k}|D_{t-1}, s_t) - \overline{\text{alg}}_E(a_{t,k}|D_{t-1}, s_t)\right| \cdot |r_{t,k}|$$
$$\leqslant \sum_{t=1}^{T} \sum_{k=1}^{A} (e^\varepsilon - 1)$$
$$= (e^\varepsilon - 1)AT.$$

Finally, since $\varepsilon > 0$ is small, we use the approximation

$$e^\varepsilon = 1 + \varepsilon + \mathcal{O}(\varepsilon^2),$$

and then the policy imitation error becomes $\varepsilon AT$.

Next, we leverage the result from Lin et al. (2024):

$$\left|\mathfrak{R}_{\text{alg}_{\hat{\theta}}}(T) - \mathfrak{R}_{\overline{\text{alg}}_B}(T)\right| = \mathcal{O}\left(T^2 \sqrt{\mathcal{R}_{\text{alg}_{\hat{\theta}}, \overline{\text{alg}}_B}}\left(\sqrt{\frac{\log(\mathcal{N}_\Theta T/\delta)}{N}} + \sqrt{\varepsilon_{\text{real}}}\right)\right) \qquad (15)$$

where

$$\log \mathbb{E}_{\overline{D}_T \sim \mathbb{P}_{\text{alg}_0, \text{alg}_B}}\left[\frac{\overline{\text{alg}}_B(\overline{a}_t|D_{t-1}, s_t)}{\text{alg}_{\hat{\theta}}(\overline{a}_t|D_{t-1}, s_t)}\right] \leqslant \varepsilon_{\text{real}}$$

for all $t \in [T]$. We extend this result to non-stationary settings to derive a bound for $|\mathfrak{R}_{\text{alg}_{\hat{\theta}}}(T) - \mathfrak{R}_{\overline{\text{alg}}_E}(T)|$.

Suppose there are $n_0 + 1$ orders of instances, and $\text{alg}_B$ runs for $T$ rounds. Let $k_i \geqslant 0$ represent the number of rounds for order-$i$ instances, so that $k_0 + k_1 + \cdots + k_{n_0} = T$. Define $T_i$ as:

$$T_i := \{t_1^{(i)}, ..., t_{k_i}^{(i)}\}, \quad |T_i| = k_i, \quad i \in \{0, ..., n_0\}$$

which is the set of rounds during which order-$i$ instances are active. If $k_i = 0$, order-$i$ instances are inactive. If $k_i > 0$, one or more order-$i$ instances are active. Using Lemma 7, we know that running one instance during $T_i$ yields a higher regret bound (as in (15)) than running zero or multiple

instances in $T_i$. Furthermore, Theorem 6 guarantees that the regret for a reinforcement learning algorithm at $T_i$ is bounded by $\varepsilon A k_i$. Thus, we have:

$$\left| \mathfrak{R}_{\mathsf{alg}_{\hat{\theta}}}(T_i) - \mathfrak{R}_{\overline{\mathsf{alg}}_E}(T_i) \right| = \mathcal{O}\left( k_i^2 \sqrt{\mathcal{R}_{\mathsf{alg}_{\hat{\theta}}, \overline{\mathsf{alg}}_B}} \left( \sqrt{\frac{\log(\mathcal{N}_\Theta k_i/\delta)}{N}} + \sqrt{\varepsilon_{\mathrm{real}}} \right) + \varepsilon A k_i \right).$$

Summing over all $T_i$, we get:

$$
\begin{aligned}
\left| \mathfrak{R}_{\mathsf{alg}_{\hat{\theta}}}(T) - \mathfrak{R}_{\overline{\mathsf{alg}}_E}(T) \right| &\leqslant \sum_{i=0}^{n_0} \left| \mathfrak{R}_{\overline{\mathsf{alg}}_{\hat{\theta}}}(T_i) - \mathfrak{R}_{\mathsf{alg}_E}(T_i) \right| \\
&\leqslant \sum_{i=0}^{n_0} \left| R_{\mathsf{alg}_{\hat{\theta}}}(T_i) - R_{\overline{\mathsf{alg}}_E}(T_i) \right| \\
&= \mathcal{O}\left( \sum_{i=0}^{n_0} k_i^2 \sqrt{\mathcal{R}_{\mathsf{alg}_{\hat{\theta}}, \overline{\mathsf{alg}}_B}} \left( \sqrt{\frac{\log(\mathcal{N}_\Theta k_i/\delta)}{N}} + \sqrt{\varepsilon_{\mathrm{real}}} \right) + \varepsilon A k_i \right) \\
&\leqslant \mathcal{O}\left( \sum_{i=0}^{n_0} k_i^2 \sqrt{\mathcal{R}_{\mathsf{alg}_{\hat{\theta}}, \overline{\mathsf{alg}}_B}} \left( \sqrt{\frac{\log(\mathcal{N}_\Theta T/\delta)}{N}} + \sqrt{\varepsilon_{\mathrm{real}}} \right) + \varepsilon A k_i \right) \\
&\leqslant \mathcal{O}\left( \left(\sum_{i=0}^{n_0} k_i\right)^2 \sqrt{\mathcal{R}_{\mathsf{alg}_{\hat{\theta}}, \overline{\mathsf{alg}}_B}} \left( \sqrt{\frac{\log(\mathcal{N}_\Theta T/\delta)}{N}} + \sqrt{\varepsilon_{\mathrm{real}}} \right) + \sum_{i=0}^{n_0} \varepsilon A k_i \right) \\
&= \mathcal{O}\left( T^2 \sqrt{\mathcal{R}_{\mathsf{alg}_{\hat{\theta}}, \overline{\mathsf{alg}}_B}} \left( \sqrt{\frac{\log(\mathcal{N}_\Theta T/\delta)}{N}} + \sqrt{\varepsilon_{\mathrm{real}}} \right) + \varepsilon A T \right),
\end{aligned}
$$

where the final inequality follows from the Cauchy–Schwarz inequality.

By Assumption 2, we have

$$\frac{\overline{\mathsf{alg}}_B(\overline{a}_t | D_{t-1}, s_t)}{\mathsf{alg}_{\hat{\theta}}(\overline{a}_t | D_{t-1}, s_t)} < e^\varepsilon.$$

Therefore,

$$\log \mathbb{E}_{\overline{D}_T \sim \mathbb{P}_{\mathsf{alg}_0, \mathsf{alg}_B}} \left[ \frac{\overline{\mathsf{alg}}_B(\overline{a}_t | D_{t-1}, s_t)}{\mathsf{alg}_{\hat{\theta}}(\overline{a}_t | D_{t-1}, s_t)} \right] \leqslant \varepsilon_{\mathrm{real}} \leqslant \varepsilon.$$

Combining this with the result of Theorem 6 completes the proof. $\qquad\square$

### D.5.1 An auxiliary lemma

**Lemma 7** (Regret bound for multiple instances)**.** *Let* $a, b, c, N, C_1, C_2 > 0$ *with* $a + b = c$. *The following bound holds:*

$$a^2 \left( \sqrt{\frac{\log(C_1 a)}{N}} + C_2 \right) + b^2 \left( \sqrt{\frac{\log(C_1 b)}{N}} + C_2 \right) \leqslant c^2 \left( \sqrt{\frac{\log(C_1 c)}{N}} + C_2 \right). \quad (16)$$

**Proof of Lemma 7.** Let $0 < u < 1$ be such that $a = uc$ and $b = (1-u)c$. Substituting into (16), we obtain:

$$u^2 \left( \sqrt{\frac{\log(C_1 u c)}{N}} + C_2 \right) + (1-u)^2 \left( \sqrt{\frac{\log(C_1 (1-u) c)}{N}} + C_2 \right) \leqslant c^2 \left( \sqrt{\frac{\log(C_1 c)}{N}} + C_2 \right).$$

Denoting $G := \log(C_1 c)/N$, we rewrite the inequality as:

$$u^2 \left( \sqrt{\log u + G} + C_2 \right) + (1-u)^2 \left( \sqrt{\log(1-u) + G} + C_2 \right) \leqslant \sqrt{G} + C_2. \quad (17)$$

Since $0 < u < 1$, we have $\log u < 0$ and $\log(1-u) < 0$. Substituting these into (17), we obtain:

$$u^2 + (1-u)^2 \leqslant 1,$$

which follows directly from $0 < u < 1$, as $u^2 + (1-u)^2 = 1 - 2u(1-u) < 1$. $\qquad\square$

# E Proofs in Section 4

## E.1 Structural Approximation

To match the order-$n$ instance length as described in Appendix C, we consider a single input matrix $\mathbf{H} = [\mathbf{h}_1, ..., \mathbf{h}_{2^n}] \in \mathbb{R}^{d \times 2^n}$, where $\{\mathbf{h}_i\}_{i=1}^{2^n} \subset \mathbb{R}^d$. This matrix is then extended to $\mathcal{H} \in \mathbb{R}^{D \times 2^n}$ defined as $\mathcal{H} := \left[\mathbf{H}^{(0)}; \ldots; \mathbf{H}^{(n)}; \mathbf{H}^*\right] \in \mathbb{R}^{D \times 2^n}$ (Figure 8) where $D := ((d+5)(n+1) + 5)$, and $\mathbf{H}^*$ contains auxiliary entries for approximation purposes. Here, $\mathbf{H}^{(i)} := [\mathbf{h}_1^{(i)}, \ldots, \mathbf{h}_{2^n}^{(i)}]$ and $\mathbf{h}_t^{(i)} = [\mathbf{h}_t; 2^i, 0, 0, 0, 0]$ where the four auxiliary entries capture the order information. Details of each entry are given in (21) and (22). Once $\mathcal{H}$ is defined, the transformer produces the output $\overline{\mathcal{H}} \in \mathbb{R}^{D \times 2^n}$. Note that even if the input length $T$ is not exactly $2^n$, the transformer can still replicate the operations of MASTER and generate a length-$T$ output by selecting the smallest $2^n$ greater than $T$ and running the algorithm for $T$ rounds.

We then define functions $\sigma_1$ and $\sigma_2$ to replicate the MALG operation: $\sigma_1$ stochastically schedules instances for each order, and $\sigma_2$ selects the instance with the lowest order to remain active in each round.

**Definition 4** ($\sigma_1$). *Given a non-increasing function $\rho : [2^n] \to \mathbb{R}$, we define $\sigma_1^\rho : \mathbb{R}^{D \times 2^n} \to \mathbb{R}^{D \times 2^n}$ as*

$$\sigma_1^\rho(\mathcal{H}) = \left[\sigma_1(\mathbf{H}^{(0)}, 0, n, \rho); \cdots ; \sigma_1(\mathbf{H}^{(n)}, n, n, \rho)\right],$$

*where for each $i \in \{0, ..., n\}$*

$$\sigma_1(\mathbf{H}^{(i)}, i, n, \rho) = \left\{\mathbf{h}_j^{(i)} \cdot \mathrm{B}\left(\frac{\rho(2^n)}{\rho(2^i)}\right) \,\middle|\, j \in \{t, t+1, ..., t+2^i-1\}, \ t \bmod 2^i = 1\right\} \in \mathbb{R}^{D \times 2^n}.$$

*Here, $\mathrm{B}(k)$ is a Bernoulli random variable with parameter $k$, meaning $\mathrm{B}(k) = 1$ with probability $k \leqslant 1$ and $\mathrm{B}(k) = 0$ otherwise.*

Using $\sigma_1$, multiple instances can be scheduled simultaneously (represented as the bluish blocks in Figure 8). Since only one instance can be active the environment at any given time, we define $\sigma_2$ to select the instance with the lowest order at each time step $t$.

We denote $\mathfrak{h}_t := \left[\mathbf{h}_t^{(0)}; \ldots; \mathbf{h}_t^{(n)}\right] \in \mathbb{R}^D$ for each $t \in [2^n]$, so that $\mathcal{H} = [\mathfrak{h}_1, \ldots, \mathfrak{h}_{2^n}]$. Then, $\sigma_2$ is defined as follows.

**Definition 5** ($\sigma_2$). *We define $\sigma_2 : \mathbb{R}^{D \times 2^n} \to \mathbb{R}^{D \times 2^n}$ by*

$$\sigma_2(\mathcal{H}) = \left[\mathfrak{h}_1', ..., \mathfrak{h}_{2^n}'\right],$$

$$\mathfrak{h}_t' = \left[\mathbf{0}; ...; \mathbf{0}; \mathbf{h}_t^{(k)}; \mathbf{0}; ...; \mathbf{0}; *; *; *; *; *\right] \ \left(\mathbf{h}_t^{(0)}, ..., \mathbf{h}_t^{(k-1)} = \mathbf{0}, \ \mathbf{h}_t^{(k)} \neq \mathbf{0}, \ 0 \leqslant k \leqslant n\right)$$

*to reproduce the uniqueness of an instance scheduled at every moment in MALG. Here, $*$s denote the last five entries of $\mathfrak{h}_t$.*

After applying $\sigma_2$, only a single instance remains active at each time $t$; this active instance is always the one with the lowest order among all scheduled instances. The reddish blocks in Figure 8 illustrate this selection.

To approximate the stationary tests (**Test 1** and **Test 2**) in Algorithm 2, we define the following:

**Definition 6** (TEST). *TEST represents the stationary tests in MASTER, involving two functions* test1 *and* test2 *that map from $\mathbb{R}^D \times \mathbb{R}^{D \times 2^n}$ to $\mathbb{R}$. Given $\widehat{\rho}$ defined in Algorithm 2 and an active instance in $\mathfrak{h}_t$ of order $m$, where $\mathfrak{h}_t$ contains $m, t, U_t, \widetilde{r}_t$, and $r_t$, we define:*

$$\texttt{test1}_{\widehat{\rho}}(\mathfrak{h}_t, \mathcal{H}) = \begin{cases} 1, & \text{if some order} - m \ \texttt{alg} \text{ ends and } \frac{1}{2^m} \sum_{\tau=t-2^m+1}^{t} r_\tau < U_t + 9\widehat{\rho}(2^m) \\ 0, & \text{otherwise} \end{cases},$$

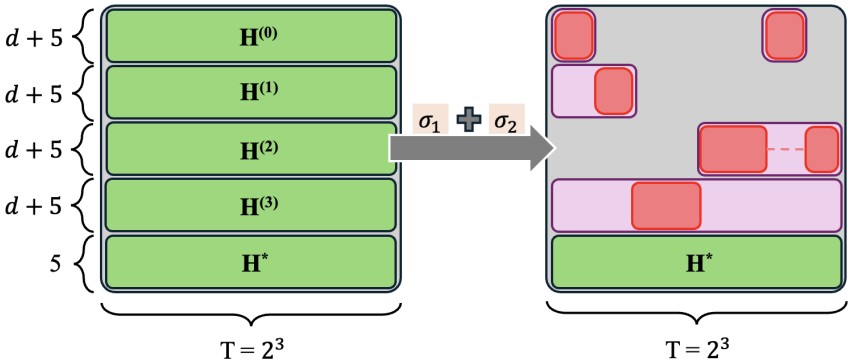

Figure 8: Illustration of $\mathcal{H}$ when $n = 3$. Purplish blocks represent instances scheduled by $\sigma_1$, while reddish blocks represent the active instances selected by $\sigma_2$. Reddish blocks connected by a dashed line are concatenated.

$$\text{test2}_{\widehat{\rho}}(\mathfrak{h}_t, \mathcal{H}) = \begin{cases} 1, & \text{if } \frac{1}{t - t_n + 1} \sum_{\tau=1}^{t} (\widetilde{r}_\tau - r_\tau) < 3\widehat{\rho}(t - t_n + 1) \\ 0, & \text{otherwise} \end{cases}.$$

TEST *is then defined as:*

$$\text{TEST}_{\widehat{\rho}}(\mathfrak{h}_t, \mathcal{H}) := \text{test1}(\mathfrak{h}_t, \mathcal{H}) \cdot \text{test2}(\mathfrak{h}_t, \mathcal{H}) = \begin{cases} 1, & \text{if } \text{test1}(\mathfrak{h}_t) = \text{test2}(\mathfrak{h}_t) = 1 \\ 0, & \text{otherwise} \end{cases}.$$

Finally, we define the noncontinuous transformer. This transformer incorporates the stationarity tests and halts processing of the remaining sequence whenever a test fails. To implement this, we design a transformer module that processes each element individually, performs the required tests, and then concatenates the results.

**Definition 7** (Noncontinuous Transformer). *For $t \in [2^n]$, denote*

$$\mathfrak{h}_t^* = \left( \text{MLP}_{\theta_{mlp}^{(L)}} \circ \text{Attn}_{\theta_{attn}^{(L)}} \right) \circ \cdots \circ \left( \text{MLP}_{\theta_{mlp}^{(1)}} \circ \text{Attn}_{\theta_{attn}^{(1)}} \right) (\mathfrak{h}_t, \mathcal{H}) \in \mathbb{R}^D,$$

*where the parameters are $\theta_{attn}^{(\ell)} = \{\mathbf{V}_m^{(\ell)}, \mathbf{Q}_m^{(\ell)}, \mathbf{K}_m^{(\ell)}\}_{m \in [2^n]} \subset \mathbb{R}^{D \times D}$ and $\theta_{mlp}^{(\ell)} = \{(\mathbf{W}_1^{(\ell)}, \mathbf{W}_2^{(\ell)})\}_{m \in [2^n]} \subset \mathbb{R}^{D' \times D} \times \mathbb{R}^{D \times D'}$. After processing $\mathfrak{h}_t$, the model interacts with the environment, observes the reward, and inserts it into the sequence: $\mathfrak{h}_t^* \to \widetilde{\mathfrak{h}}_t$ ((21) and (22)). By concatenating $\widetilde{\mathfrak{h}}_{t \in [2^n]}$, we define the L-layer noncontinuous transformer $\mathcal{TF}_\theta : \mathbb{R}^{D \times 2^n} \to \mathbb{R}^{D \times 2^n}$ as*

$$\mathcal{TF}_\theta = \mathcal{TF}_\theta^{(2^n)} \tag{18}$$

$$\mathcal{TF}_\theta^{(T)}(\mathcal{H}) = \widetilde{\mathcal{H}}^{(T)} \in \mathbb{R}^{D \times 2^n} \quad (T \in [2^n]) \tag{19}$$

*where $\widetilde{\mathcal{H}}^{(T)} := \left[ \widetilde{\mathfrak{h}}_1, ..., \widetilde{\mathfrak{h}}_T, \mathbf{T}_T \mathfrak{h}_{T+1}, ..., \mathbf{T}_T \mathfrak{h}_{2^n} \right] \in \mathbb{R}^{D \times 2^n}$. Here, $\mathbf{T}_T \in \mathbb{R}^{D \times D}$ is the test matrix at time $T$:*

$$\mathbf{T}_T := \text{diag}(\underbrace{1, ..., 1}_{d+1}, \overbrace{\text{TEST}, \text{TEST}, \text{TEST}, \text{TEST}}^{\times (n+1)}, 1, 1, \text{TEST}, \text{TEST}, \text{TEST}) \tag{20}$$

*where $\text{TEST} = \text{TEST}(\overline{\mathfrak{h}}_T, \mathcal{H}^{(T)})$. Order information $\sum_{\tau=t-2^i+1}^{t-1} r_\tau, \sum_{k=0}^{i-1} \widetilde{r}_t^{(k)}, \widetilde{r}_t^{(i)}$, historical data $\sum_{\tau=1}^{t-1} r_\tau, \sum_{\tau=1}^{t-1} \widetilde{r}_\tau, U_{t-1}$, and rand in (21) and (22) are reset when $\text{TEST} = 0$. $\widetilde{r}_t^{(i)}$ denotes the auxiliary value generated at order $i$.*

**Interaction with the environment**   To illustrate the transformer's interaction with the environment, we consider the following:

$$
\mathbf{h}_t^{(i)} = \begin{bmatrix} \mathbf{x}_t^{(i)} \\ 2^i \\ \sum_{\tau=t-2^i+1}^{t-1} r_\tau \\ 0 \\ \sum_{k=0}^{i-1} \widetilde{r}_t^{(k)} \\ \widetilde{r}_t^{(i)} \end{bmatrix} \xrightarrow[\text{Interact with the environment}]{\text{Processed by the module}} \begin{bmatrix} \overline{\mathbf{x}}_t^{(i)} \\ 2^i \\ \sum_{\tau=t-2^i+1}^{t} r_\tau \\ \texttt{rand} \\ \sum_{k=0}^{i-1} \widetilde{r}_t^{(k)} \\ \widetilde{r}_t^{(i)} \end{bmatrix} = \overline{\mathbf{h}}_t^{(i)}, \qquad (21)
$$

$$
\mathfrak{h}_t = \begin{bmatrix} \mathbf{h}_t^{(0)} \\ \vdots \\ \mathbf{h}_t^{(n)} \\ 2^n \\ t \\ \sum_{\tau=1}^{t-1} r_\tau \\ \sum_{\tau=1}^{t-1} \widetilde{r}_\tau \\ U_{t-1} \end{bmatrix} \xrightarrow[\text{Interact with the environment}]{\text{Processed by the module}} \begin{bmatrix} \overline{\mathbf{h}}_t^{(0)} \\ \vdots \\ \overline{\mathbf{h}}_t^{(n)} \\ 2^n \\ t \\ \sum_{\tau=1}^{t} r_\tau \\ \sum_{\tau=1}^{t} \widetilde{r}_\tau \\ U_t \end{bmatrix} = \overline{\mathfrak{h}}_t. \qquad (22)
$$

Here, $\sum_{\tau=1}^{0} r_\tau = 0$ and $r_\tau = 0$ for $\tau \leqslant 0$. The $\texttt{rand}$ entry is a random number generated by the transformer. The vectors $\mathbf{x}_t^{(i)} \in \mathbb{R}^d$ are used for in-context learning, and $\overline{\mathbf{x}}_t^{(i)}$ denotes their processed forms; for example, these vectors can encode action choices in bandit problems (Lin et al., 2024).

Each entry of the vectors captures cumulative information or constants relevant to the model's operations up to time $t$. After passing the sequence through the MLP+Attention module, certain orders are scheduled at each time $t$, while all other instances are set to $\mathbf{0}$. The $\sigma_2$ operation ensures that only one instance remains active at a time by selecting the instance with the lowest order and zeroing out all others. If the active instance is order $q \leqslant n$, then $\widetilde{r}_t^{(q)}$ is assigned as $\widetilde{r}_t$ at time $t$, and the update $U_t = \min\{U_{t-1}, \widetilde{r}_t^{(q)}\}$ is applied. Finally, the module interacts with the environment, observes the reward $R_t$, and updates the tensor accordingly.

**Notation with regard to the input sequence**   To avoid potential confusion, the notations defined above are summarized below:

- $\mathbf{h}_t \in \mathbb{R}^d$: The $t$-th element of the input sequence.
- $\mathbf{H} = [\mathbf{h}_1, \ldots, \mathbf{h}_{2^n}] \in \mathbb{R}^{d \times 2^n}$: The input sequence as a matrix.
- $\mathcal{H} = [\mathbf{H}^{(0)}; \cdots; \mathbf{H}^{(n)}; \mathbf{H}^*] = [\mathfrak{h}_1 \cdots \mathfrak{h}_{2^n}] \in \mathbb{R}^{D \times 2^n}$: The extended input sequence, where $\mathbf{H}^{(i)}$ denotes the order-$i$ sequence.
- $\mathfrak{h}_t = \left[\mathbf{h}_t^{(0)}; \cdots; \mathbf{h}_t^{(n)}; \mathbf{h}_t^*\right] \in \mathbb{R}^D$: The $t$-th vector of the extended sequence.
- $\mathfrak{h}_t' = \left[\mathbf{0}; \cdots; \mathbf{0}; \mathbf{h}_t^{(k)}; \mathbf{0}; \cdots; \mathbf{0}; \mathbf{h}_t^*\right] \in \mathbb{R}^D$: The $t$-th vector of the extended input after being processed by $\sigma_2$. Here, $k$ denotes the order of the active instance selected by $\sigma_2$, i.e., the lowest nonzero order at time $t$.
- $\mathfrak{h}_t^*$: The result of processing $\mathfrak{h}_t$ by the noncontinuous transformer module.
- $\overline{\mathfrak{h}}_t$: The output of the noncontinuous transformer after incorporating the observed reward at time $t$.

**Restart**   The restart mechanism (line 9 of Algorithm 2) is integrated into the noncontinuous transformer. If all stationary tests pass, the matrices $\mathbf{T}_T$ ($T \in [2^n]$) remain identity matrices, and the sequence is processed normally. If a test fails at time $T$, the matrices $\mathbf{T}_T$ ($T \in [2^n]$) modify the states $\overline{\mathfrak{h}}_{T+1}, \ldots, \overline{\mathfrak{h}}_{2^n}$: the first $D - 7$ elements—including $\mathbf{x}_T$, $T$, $2^n$, and $2^i$—are preserved, while the remaining seven elements are set to zero. This ensures that when a test fails, all historical information is erased, but essential identifiers and order information are retained to start a new block.

**Rollout** The noncontinuous transformer's architecture is more complex than the classic version, so we outline its input-output flow.

Starting with an input sequence $\mathbf{H} \in \mathbb{R}^{D \times 2^n}$, we extend it to $\mathcal{H} = [\mathbf{H}^{(0)}; \cdots; \mathbf{H}^{(n)}; \mathbf{H}^*] \in \mathbb{R}^{D \times 2^n}$, where submatrices $\mathbf{H}^{(i)}$ and $\mathbf{H}^{(j)}$ are only different in several entries to record order information.

$\mathcal{H}$ is passed through $\sigma_1$ and $\sigma_2$: $\sigma_1$ schedules instances for each order, and $\sigma_2$ activates only the instance with the lowest order at any given time $t$. As a result, at each time $t$, only one instance is active, and the active instance may come from a different order at each time step.

Next, each element in the sequence is processed by the traditional MLP+Attention module. The module uses this information to interact with the environment, collects the reward, and inserts it into the sequence: $\mathfrak{h}_t^* \to \overline{\mathfrak{h}}_t$ ((21) and (22)). The updated sequence $\overline{\mathfrak{h}}_t$ is evaluated by TEST, which performs stationary tests. If the tests pass, the block remains unchanged; otherwise, the remaining block entries are set to zero, keeping only essential information. This effectively ends the current $2^n$-length block and starts a new $2^n$-length block (line 2, Algorithm 2). On restart, variables like $R_t$ and $f_t$ are reinserted at their appropriate positions, following the same procedure. After processing the entire sequence, the final tensor $\overline{\mathcal{H}}$ is produced. By gathering the rewards from all orders at each $t$ (only one reward is nonzero at a given time), the resulting sequence $\overline{\mathbf{H}}$ serves as the output of the noncontinuous transformer, containing cumulative rewards up to time $2^n$.

### E.2 Motivation of the Non-Continuous Transformer

In this section, we first briefly demonstrate how the proofs can be adapted to a regular transformer, then explain the motivation of using the non-continuous transformer.

The core challenge in using a regular transformer lies in approximating $\sigma_2$ (Definition 5), which must select the first non-zero entry among all entries. This operation requires comparing all trajectories simultaneously to identify the first non-zero one, and it is the main reason why the augmented inputs are used.

One alternative approach is to use a regular transformer with $n + 1$ heads to approximate $\sigma_1$ and $\sigma_2$ as follows:

- Layer 1 (**Compression**): Each head uses masked multi-head attention to extract per-trajectory representations, i.e., $\mathbf{H}^{(i)} = \mathbf{W}_i^{\text{comp}} \cdot \text{MaskAttn}_m(\mathbf{H}) \in \mathbb{R}^{\frac{D}{n+1} \times T}$. Here, MaskAttn restricts information flows across heads, and the combined output is $\mathbf{H}' = [\mathbf{H}^{(0)}; ...; \mathbf{H}^{(n)}] \in \mathbb{R}^{D \times T}$.

- Layer 2~K: Generate Bernoulli masks, block masks, and select the first non-zero trajectory to approximate $\sigma_1$ and $\sigma_2$ (same as the procedure in Appendix E.3.2 and E.3.3).

- Final Layer (**Decompression**): Reconstruct the selected trajectory: $\mathbf{H}'' = \sum_{m=0}^{n} \mathbf{W}_m^{\text{decomp}} \cdot \mathbf{H}^{(m)} \in \mathbb{R}^{D \times T}$.

However, this method adds several analytical burdens, such as ensuring that $\mathbf{W}_m^{\text{decomp}} \cdot \mathbf{W}_m^{\text{comp}} \approx I_D$ for $m \in \{0, ..., n\}$ to preserve information, and constructing MaskAttn, which make the analysis more cumbersome.

In contrast, our proposed method is a simple and transparent preprocessing step where augmented inputs are generated via a linear transformation $\mathfrak{h}_t = \mathbf{M}_t \cdot \mathbf{h}_t$. This preserves the core semantics while avoiding unnecessary architectural detours. Importantly, the resulting non-continuous transformer is **structurally equivalent** to a standard transformer with transformed inputs.

### E.3 Approximation of WS (Proposition 3)

We divide the proof into three parts: approximating random number generation, $\sigma_1$, and $\sigma_2$. Throughout this section, we denote the ReLU activation function by $\sigma(\cdot)$.

### E.3.1 Approximating random number generation

We start by approximating $\mathbb{1}_{>0}[\cdot]$. According to the definition of the ReLU function, we have

$$\sigma(kx) - \sigma(kx - 1) = \begin{cases} 0, & x \leqslant 0 \\ kx, & 0 < x \leqslant \frac{1}{k} \\ 1, & x > \frac{1}{k} \end{cases} \quad .$$

When $k \to \infty$, it approximates $\mathbb{1}[x] = \begin{cases} 0, & x \leqslant 0 \\ 1, & x > 0 \end{cases}$ .

We denote $z_k := \mathbf{x}_{k,0}$ and $P_z(z)$ being the cumulative distribution function (CDF) of $\{z_1, ..., z_{2^n}\}$. Following the proof in Hataya and Imaizumi (2024), we construct a 2-head attention layer to approximate the CDF.

From the approximation of $\mathbb{1}[\cdot]$, $P_z(t) = 1/2^n \sum_{k=1}^{2^n} \mathbb{1}[\mathbf{x}_{k,0} \leqslant t]$ can be approximated by sum of ReLU functions as

$$\widehat{P}_z(t) \approx \frac{1}{2^n} \sum_{k=1}^{2^n} \{\sigma(k(t - x)) - \sigma(k(t - x) - 1)\}$$

where $k$ is sufficiently large. Suppose the vector in formula 21 and 22 (left) is one of the input $\mathbf{h}_t$. By selecting $\{\mathbf{Q}_{1,2}, \mathbf{K}_{1,2}, \mathbf{V}_{1,2}\}$ such that

$$\mathbf{Q}_1 \mathbf{h}_i = \begin{bmatrix} k\mathbf{x}_{i,0} \\ -1 \\ 0 \\ \mathbf{0} \end{bmatrix}, \quad \mathbf{Q}_2 \mathbf{h}_i = \begin{bmatrix} k\mathbf{x}_{i,0} \\ -1 \\ -1 \\ \mathbf{0} \end{bmatrix}, \quad \mathbf{K}_1 \mathbf{h}_i = \mathbf{K}_2 \mathbf{h}_i = \begin{bmatrix} 1 \\ k\mathbf{x}_{i,0} \\ 1 \\ \mathbf{0} \end{bmatrix},$$

$$\mathbf{V}_1 \mathbf{h}_i = \begin{bmatrix} 1 \\ \mathbf{0} \end{bmatrix}, \quad \mathbf{V}_2 \mathbf{h}_i = \begin{bmatrix} -1 \\ \mathbf{0} \end{bmatrix},$$

we have

$$\sum_{m=1}^{2} \sigma\left(\langle \mathbf{Q}_m \mathbf{h}_j, \mathbf{K}_m \mathbf{h}_i \rangle\right) \mathbf{V}_m \mathbf{h}_i = \sigma\left(k(\mathbf{x}_{j,0} - \mathbf{x}_{i,0})\right) - \sigma\left(k(\mathbf{x}_{j,0} - \mathbf{x}_{i,0}) - 1\right).$$

Therefore, the output is

$$\widetilde{\mathbf{h}}_i = \mathbf{h}_i + \frac{1}{2^n} \sum_{j=1}^{2^n} \sum_{m=1}^{2} \sigma\left(\langle \mathbf{Q}_m \mathbf{h}_i, \mathbf{K}_m \mathbf{h}_i \rangle\right) \mathbf{V}_m \mathbf{h}_i = \begin{bmatrix} \texttt{rand} \\ \vdots \end{bmatrix}$$

where $\texttt{rand} = \widehat{P}_z(\mathbf{x}_{i,0})$ can be regarded as a random variable sampled from $\mathcal{U}(0, 1)$. $\qquad \square$

### E.3.2 Approximating $\sigma_1$

**Step 1. Generating Bernoulli masks.** Define Bernoulli Masks as $\mathtt{B}_{i,t} = \mathbb{1}[\texttt{rand}_{i,t} \leqslant \rho(2^n)/\rho(2^i)]$. We consider a random number $\texttt{rand}_{i,t} \in \mathbb{R}$ at order $i$ and time $t$, and $\texttt{rand}_{i,t}, \rho(\cdot) \in [0, 1]$. Following the approach in Step 1, we begin by considering the input $\mathbf{h}_t^{(i)}$. This input includes $\texttt{rand}_{i,t}, 2^i, 2^n$, and we also assume that $\rho(2^i)$ and $\rho(2^n)$ are added to $\mathbf{h}_t^{(i)}$ through $\rho(\cdot)$. In this case, a two-layer MLP can implement Bernoulli Masks.

First, consider the function $f(x, y) = x \cdot y \in \mathbb{R}$. For $x, y \in [0, 1]$, dividing the domain $[0, 1] \times [0, 1]$ into $\lfloor 1/\epsilon \rfloor$ segments allows us to express it as:

$$\mathbf{W} \begin{bmatrix} x \\ y \end{bmatrix} + \mathbf{b} = \sum_{k,l=1}^{n} \alpha_{k,l} x + \beta_{k,l} y + \gamma_{k,l}.$$

This form can approximate $f$ using an MLP. Based on this strategy, the two-layer MLP is constructed as follows:

$$z_1 = \sigma(\mathbf{W}_1 \mathbf{h}_t^{(i)} + \mathbf{b}_1)$$

$$
= \sigma\left(\begin{bmatrix}
k_2\left(\rho(2^n) - (\alpha_{1,1}\mathtt{rand}_{i,t} + \beta_{1,1}\rho(2^i) + \gamma_{1,1})\right) \\
\vdots \\
k_2\left(\rho(2^n) - (\alpha_{\lfloor\frac{1}{\sqrt{\varepsilon}}\rfloor,\lfloor\frac{1}{\sqrt{\varepsilon}}\rfloor}\mathtt{rand}_{i,t} + \beta_{\lfloor\frac{1}{\sqrt{\varepsilon}}\rfloor,\lfloor\frac{1}{\sqrt{\varepsilon}}\rfloor}\rho(2^i) + \gamma_{\lfloor\frac{1}{\sqrt{\varepsilon}}\rfloor,\lfloor\frac{1}{\sqrt{\varepsilon}}\rfloor})\right) \\
k_2\left(\rho(2^n) - (\alpha_{1,1}\mathtt{rand}_{i,t} + \beta_{1,1}\rho(2^i) + \gamma_{1,1})\right) - 1 \\
\vdots \\
k_2\left(\rho(2^n) - (\alpha_{\lfloor\frac{1}{\sqrt{\varepsilon}}\rfloor,\lfloor\frac{1}{\sqrt{\varepsilon}}\rfloor}\mathtt{rand}_{i,t} + \beta_{\lfloor\frac{1}{\sqrt{\varepsilon}}\rfloor,\lfloor\frac{1}{\sqrt{\varepsilon}}\rfloor}\rho(2^i) + \gamma_{\lfloor\frac{1}{\sqrt{\varepsilon}}\rfloor,\lfloor\frac{1}{\sqrt{\varepsilon}}\rfloor})\right) - 1
\end{bmatrix}\right),
$$

where $\mathbf{W}_1 \in \mathbb{R}^{2\lfloor 1/\varepsilon\rfloor \times D}$. The output is then processed by another layer:

$$
\mathbf{W}_2 z_1 + \mathbf{b}_2 = \sigma\left(k_2\left(\rho(2^n) - \sum_{k,l=1}^{n} (\alpha_{k,l}\mathtt{rand}_{i,t} + \beta_{k,l}\rho(2^i) + \gamma_{k,l})\right)\right)
$$

$$
- \sigma\left(k_2\left(\rho(2^n) - \sum_{k,l=1}^{n} (\alpha_{k,l}\mathtt{rand}_{i,t} + \beta_{k,l}\rho(2^i) + \gamma_{k,l})\right) - 1\right)
$$

$$
\approx \sigma\left(k_2(\rho(2^n) - \mathtt{rand}\cdot\rho(2^i))\right) - \sigma\left(k_2(\rho(2^n) - \mathtt{rand}\cdot\rho(2^i)) - 1\right)
$$

$$
\longrightarrow \begin{cases} 1, & \mathtt{rand}_{i,t} \leqslant \frac{\rho(2^n)}{\rho(2^i)}, \\ 0, & \mathtt{rand}_{i,t} > \frac{\rho(2^n)}{\rho(2^i)}. \end{cases} \quad (k_2 \to \infty), \quad \mathbf{W}_2 \in \mathbb{R}^{1\times 2\lfloor 1/\varepsilon\rfloor}.
$$

The hidden dimensions of $\mathbf{W}_1, \mathbf{W}_1$ are $\mathcal{O}(1/\varepsilon)$. If the input is expanded from $\mathbf{h}_t^{(i)}$ to $\mathbf{H}^{(i)}$, the hidden dimensions increase to $\mathcal{O}(2^n/\varepsilon)$. Further expand further to $\mathcal{H}$, the hidden dimensions of $\mathbf{W}_1, \mathbf{W}_1$ become $\mathcal{O}(n2^n/\varepsilon) = \mathcal{O}(T\log_2 T/\varepsilon)$. The operator norms $\|\mathbf{W}_1\|_{\mathrm{op}}, \|\mathbf{W}_1\|_{\mathrm{op}}$ scale as $\mathcal{O}(\sqrt{T\log_2 T}/\varepsilon)$. Finally, by taking $k_2 \to \infty$, the error approaches zero.

**Step 2: Generating block masks.** By setting $\mathtt{rand}_{i,t} = \cdots = \mathtt{rand}_{i,t+2^i-1}$ $(t \equiv 1\,(mod\,2))$, we have $\mathtt{B}_{i,t} = \cdots = \mathtt{B}_{i,t+2^i-1}$ $(t \equiv 1\,(mod\,2))$. Since the output of Bernoulli masks can be expressed as $\mathcal{H} := [\mathfrak{h}_1 \cdots \mathfrak{h}_{2^n}] \in \mathbb{R}^{D\times 2^n}$, where for $t \in [2^n]$:

$$
\mathfrak{h}_t = \begin{bmatrix} \mathbf{h}_t^{(0)} \\ \vdots \\ \mathbf{h}_t^{(n)} \\ \star \end{bmatrix}, \quad
\mathbf{h}_t^{(0)} = \begin{bmatrix} \mathbf{x}_t^{(0)} \\ 0 \\ 0 \\ \mathtt{B}_{0,t} \\ 0 \\ f_t^{(0)} \end{bmatrix} \quad
\mathbf{h}_t^{(1)} = \begin{bmatrix} \mathbf{x}_t^{(1)} \\ 0 \\ 0 \\ \mathtt{B}_{1,t} \\ f_t^{(0)} \\ f_t^{(1)} \end{bmatrix}, \quad \cdots, \quad
\mathbf{h}_t^{(n)} = \begin{bmatrix} \mathbf{x}_t^{(n)} \\ 0 \\ 0 \\ \mathtt{B}_{n,t} \\ \sum_{p=0}^{n-1} f_t^{(p)} \\ f_t^{(n)} \end{bmatrix},
$$

we choose $\{\mathbf{Q}_m, \mathbf{K}_m, \mathbf{V}_m\}_{m\in\{0,\ldots,n\}}$ such that for tokens $\mathfrak{h}_t$,

$$
\mathbf{Q}_m\mathfrak{h}_t = \begin{bmatrix} t \\ 1 \\ \mathtt{B}_{m,t} \\ -1 \end{bmatrix}, \quad
\mathbf{K}_m\mathfrak{h}_k = \begin{bmatrix} -1 \\ k \\ 1 \\ 0.5 \end{bmatrix}, \quad
\mathbf{V}_m\mathfrak{h}_k = \left[\mathbf{0}; \ldots; \mathbf{0}; \mathbf{h}_k^{(m)}; \mathbf{0}; \ldots; \mathbf{0}\right]
$$

In this way, with $k < t$ we have

$$
\sum_{m=0}^{n} \mathbb{1}_{>0}\left[\langle\mathbf{Q}_m\mathfrak{h}_t, \mathbf{K}_m\mathfrak{h}_k\rangle\right]\mathbf{V}_m\mathfrak{h}_k = \mathbf{0},
$$

and with $k = t$ we have

$$\sum_{m=0}^{n} r\left(\langle \mathbf{Q}_m \mathfrak{h}_t, \mathbf{K}_m \mathfrak{h}_t \rangle\right) \mathbf{V} \mathfrak{h}_t = \sum_{m=0}^{n} \mathbb{1}_{>0} \left[\mathsf{B}_{m,t} - 0.5\right] \begin{bmatrix} \mathbf{0} \\ \vdots \\ \mathbf{0} \\ \mathbf{h}_k^{(m)} \\ \mathbf{0} \\ \vdots \\ \mathbf{0} \end{bmatrix} = \begin{bmatrix} \mathsf{B}_{0,t} \mathbf{h}_t^{(0)} \\ \vdots \\ \mathsf{B}_{n,t} \mathbf{h}_t^{(n)} \end{bmatrix}.$$

Finally, by referring to the approximation of $\mathbb{1}_{>0}[\cdot]$ (E.3.1), we note that this indicator function can be approximated using two ReLU functions. Consequently, the block mask can be approximated by a $2(n+1)$-head masked attention layer. By setting $1/\varepsilon = 4(n+1)^2$, we have $M = \mathcal{O}(1/\sqrt{\varepsilon})$

Combining this block mask with the random number generation and the Bernoulli mask completes the proof. □

### E.3.3  Approximating $\sigma_2$

Based on the output of $\sigma_1$, we can express the input of $\sigma_2$ as $\mathcal{H} := [\mathfrak{h}_1 \cdots \mathfrak{h}_{2^n}] \in \mathbb{R}^{D \times 2^n}$, where for $t \in [2^n]$:

$$\mathfrak{h}_t = \begin{bmatrix} \mathbf{h}_t^{(0)} \\ \vdots \\ \mathbf{h}_t^{(n)} \\ \star \end{bmatrix}, \quad \mathbf{h}_t^{(0)} = \begin{bmatrix} \mathbf{x}_t^{(0)} \\ 0 \\ 0 \\ 0 \\ f_t^{(0)} \end{bmatrix} \quad \mathbf{h}_t^{(1)} = \begin{bmatrix} \mathbf{x}_t^{(1)} \\ 0 \\ 0 \\ f_t^{(0)} \\ f_t^{(1)} \end{bmatrix}, \quad \ldots, \quad \mathbf{h}_t^{(n)} = \begin{bmatrix} \mathbf{x}_t^{(n)} \\ 0 \\ 0 \\ \sum_{p=0}^{n-1} f_t^{(p)} \\ f_t^{(n)} \end{bmatrix}.$$

From the definition of $\sigma_1$, we know that some $\mathbf{h}_t^{(i)}$ values are nonzero, while others are zero. If we assume $i_1 < \ldots < i_k$ $(0 < k \leq n)$ such that $\mathbf{h}_t^{(i_1)}, \ldots, \mathbf{h}_t^{(i_k)} \neq \mathbf{0}$ while all other $\mathbf{h}_t^{(n)}$ values are $\mathbf{0}$, the goal of $\sigma_2$ is to ensure that $\mathbf{h}_t^{(i_1)} \neq \mathbf{0}$ and the others being $\mathbf{0}$.

We choose $\{\mathbf{Q}_m, \mathbf{K}_m, \mathbf{V}_m\}_{m \in \{0,\ldots,n\}}$ such that for tokens $\mathfrak{h}_t$,

$$\mathbf{Q}_m \mathfrak{h}_t = \begin{bmatrix} t \\ \sum_{p=0}^{m-1} f_t^{(p)} \\ 1 \\ \epsilon_0 \end{bmatrix}, \quad \mathbf{K}_m \mathfrak{h}_k = \begin{bmatrix} -1 \\ -c_0 \\ k \\ 1 \end{bmatrix}, \quad \mathbf{V}_m \mathfrak{h}_k = \left[\mathbf{0}; \ldots; \mathbf{0}; \mathbf{h}_k^{(m)}; \mathbf{0}; \ldots, \mathbf{0}\right]$$

where $c_0 > 0$ is sufficiently large and $\epsilon_0 > 0$ is sufficiently small such that $\epsilon_0 - c_0 f_j < 0$ for any $f_j > 0$. In this way, with $k < t$ we have

$$\sum_{m=0}^{n} \mathbb{1}_{>0}\left(\langle \mathbf{Q}_m \mathfrak{h}_t, \mathbf{K}_m \mathfrak{h}_k \rangle\right) \mathbf{V}_m \mathfrak{h}_k = \mathbf{0},$$

and with $k = t$ we have

$$\sum_{m=0}^{n} r\left(\langle \mathbf{Q}_m \mathfrak{h}_t, \mathbf{K}_m \mathfrak{h}_t \rangle\right) \mathbf{V}_m \mathfrak{h}_t = \sum_{m=0}^{n} \mathbb{1}_{>0} \left[\epsilon_0 - c_0 \sum_{p=0}^{m-1} f_t^p\right] \begin{bmatrix} \mathbf{0} \\ \vdots \\ \mathbf{0} \\ \mathbf{h}_k^{(m)} \\ \mathbf{0} \\ \vdots \\ \mathbf{0} \end{bmatrix}$$

$$
= \begin{cases} \begin{bmatrix} \mathbf{0} \\ \vdots \\ \mathbf{0} \\ \mathbf{h}_k^{(m)} \\ \mathbf{0} \\ \vdots \\ \mathbf{0} \end{bmatrix}, & \text{if } \sum_{p=0}^{m-1} f_t^p = 0 \\ \mathbf{0}, & \text{otherwise} \end{cases}.
$$

Since $\sum_{p=0}^{m-1} f_t^p = 0$ is equivalent to $\mathbf{h}_t^{(0)} = \cdots = \mathbf{h}_t^{(p-1)} = \mathbf{0}$, we have for each $t \in [T]$ that

$$
\sum_{m=0}^{n} \sum_{k=1}^{t} \mathbb{1}_{>0} \left[ \langle \mathbf{Q}_m \mathfrak{h}_t, \mathbf{K}_m \mathfrak{h}_k \rangle \right] \mathbf{V}_m \mathfrak{h}_k = \begin{bmatrix} \mathbf{0} \\ \vdots \\ \mathbf{0} \\ \mathbf{h}_t^{(q)} \\ \mathbf{0} \\ \vdots \\ \mathbf{0} \end{bmatrix}
$$

where $\mathbf{h}_t^{(0)} = \cdots = \mathbf{h}_t^{(q-1)} = \mathbf{0}$ and $\mathbf{h}_t^{(m)} \neq 0$. Finally, referring the approximation of $\mathbb{1}_{>0}[\cdot]$ (E.3.1), we can approximate $\mathbb{1}_{>0}[\cdot]$ using 2 ReLU functions. Therefore, $\sigma_2$ can be approximated by a $2(n+1)$-head masked attention layer. □

### E.4 Equivalence of WS and RM with previous works

#### E.4.1 WS

Sliding windows used in previous works (e.g., (Cheung et al., 2022; Trovo et al., 2020)) can be regarded as special cases of WS. Specifically, they are equivalent to WS when WS contains only one instance, all windows are of equal size, and they are connected without overlap. Figure 9 shows the equivalence between the stochastic instance scheduler in MASTER and the sliding window schedulers from previous works.

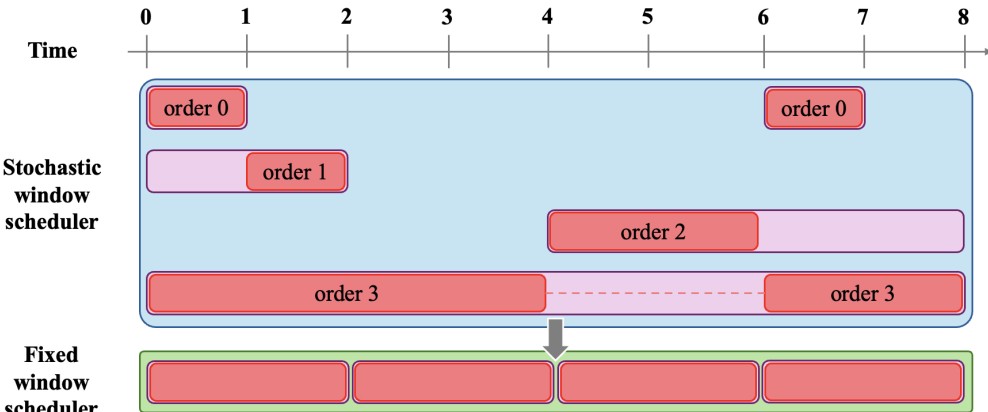

Figure 9: Equivalence between the stochastic instance scheduler in MASTER and the sliding window schedulers from previous works. Purplish blocks represent scheduled blocks, while reddish blocks represent active ones. Reddish blocks connected by a dashed line are concatenated. Here, $T = 8$ and the sliding window size is 2.

### E.4.2 `RM`

The restart strategies in previous works can be broadly categorized into two types: stochastic and deterministic (Gomes et al., 1998; Streeter and Golovin, 2008). Stochastic restarts typically involve some form of randomness or a test, whereas deterministic restarts are executed after a fixed number of rounds. Consequently, deterministic restarts can be viewed as a special case of sliding window strategies. As for stochastic restarts, given that transformers are capable of generating random numbers and MLPs can implement stationary tests, it follows from the universal approximation theorem (Definition 8) that transformers can also approximate stochastic restart mechanisms.

**Definition 8** (Universal Approximation Theorem for MLPs). *Let $f$ be a continuous function from a compact subset $K \subseteq \mathbb{R}^n$ to $\mathbb{R}^m$. Suppose $\sigma : \mathbb{R} \to \mathbb{R}$ is a non-polynomial, continuous activation function applied component-wise.*

*Then, for any $\epsilon > 0$, there exists a single hidden-layer MLP that approximates $f$ to within $\epsilon$. Specifically, there exist weights $A \in \mathbb{R}^{k \times n}$, $b \in \mathbb{R}^k$, $C \in \mathbb{R}^{m \times k}$, and a sufficiently large number of hidden units $k$ such that the MLP*

$$g(x) = C \cdot \sigma(A \cdot x + b)$$

*satisfies*

$$\sup_{x \in K} \|f(x) - g(x)\| < \epsilon.$$

