# OpenReview forum: "Optimal Dynamic Regret by Transformers for Non-Stationary Reinforcement Learning"
_NeurIPS.cc/2025/Conference — NeurIPS 2025 poster_

### Official Review · Reviewer_e7t9 · 2025-06-09

**Clarity:** 2
**Significance:** 3
**Originality:** 2
**Rating:** 4
**Confidence:** 4

**Summary:**

This paper analyzes the performance of transformers in the setting of in-context reinforcement learning (ICRL) in non-stationary environments.

The learning setup is the same as the (stationary) ICRL setup of Lin et al (2024).  The learner is given N iid trajectories of the form $D_T = (s_1, a_1, r_1, … s_T, a_T, r_T)$ generated by an offline algorithm alg0, as well as corresponding optimal actions $\bar{a_1}, …, \bar{a_T}$ generated by a base algorithm algB (which produces “high-quality” actions), the transformer is trained to maximize $ \log  p(\bar{a}_t | D_{t-1}^i, s_t^i)$.
The paper analyzes the dynamic regret of such a pre-trained transformer in a non-stationary environment, with non-stationarity defined similarly as in Wei and Luo (2021).

Similarly to Wei and Luo (2021), the authors consider the availability of a base algorithm ALG for stationary environments which produces a quantity $\tilde{r}_t$ that can be used to detect non-stationarity (such as an UCB on the reward). Then they consider the MASTER algorithm of Wei and Luo,  which converts a base algorithm ALG into one that can deal with non-stationarities (by running ALG at randomly-selected multiple time-scales, with periodic resets whenever non-stationarity is detected). The authors break down the dynamic regret of the transformer into three components: (1) regret of MASTER, (2) regret of transformer wrt MASTER, and (3) approximation error in the transformer implementing MASTER. (2) is bounded by extending the work of Lin et al (2024) to the non-stationary setting.

Bounding the approximation error is the main technical contribution. The authors show that small transformers / MLPs can be used to implement the randomized window-scheduler and restarts. They subsequently show that the overall MASTER algorithm can be implemented by a "non-continuous transformer” (only introduced and defined in the appendix).

The paper also includes some experiments on a linear bandit setting.

**Questions:**

Why is the definition of the non-stationarity measure different from Wei and Luo? They define it as the maximum change in the expected reward of a policy, $\max_\pi |r_{t+1}(\pi) - r_{t}(\pi)|$. Your definition has $\max_{s, a, s’, a’} |r_{t+1}(s’, a’) - r_t(s, a)|$ which could be non-zero for a stationary reward.

The MASTER algorithm can be implemented by the “non-continous transformer”, which is different from the standard architecture, and operates on augmented inputs. Why not state that in the main text? Does using this architecture complicate adapting the results of Lin et al (2024)?

Assumption 1 assumes that the transformer algorithm “outputs” the auxiliary quantity $\tilde{r}_t$. But the transformer is only trained to output the actions. Should it be trained to output both? Or should it be given access to some other function computing auxiliary values? Or does it suffice for the transformer to be able to compute these values in some intermediate block?

**Ethical Concerns:**

["NO or VERY MINOR ethics concerns only"]

**Final Justification:**

Minor issues resolved. Authors have committed to describing the actual architecture used for the proof into the main text.

**Limitations:**

The discussion of limitations seems inadequate. The only stated limitation is in relation to the experiments. The authors should additionally discuss the discrepancies between the analyzed non-continuous transformer and the standard architecture. The limitations discussed in Lin et al (2024) also apply to this work and could be mentioned.

**Quality:**

3

**Strengths And Weaknesses:**

Quality:
Most components of the submission seem technically sound. I am worried that the architecture and data processing described in Appendix D, used to prove Theorem 5 (that transformers can approximate the MASTER algorithm), does not match the architecture description in the main text.
The experiments are inadequately described and not especially convincing (but this is mostly a theory paper).


Clarity:

Writing is clear overall, but there is room for improvement in the organization of the paper.
In particular, the non-continuous transformer (architecture that can implement MASTER) is only introduced and defined in the appendix, and the description is difficult to parse. This architecture and data augmentation should be discussed (or at least mentioned) in the main body.

The proof outline mostly focuses on the approximation error, but it may be useful to include other components as well (e,g. Section C.2, and and outline of how the results of Lin et al are adapted to the non-stationary setting in C.5).

In the experimental setup, the first paragraph states that the action set is changing over the rounds, but the subsequent text suggests that the actions are fixed and the range of the rewards is changing. In (a) it changes to [3, 4] for some rounds (how? do you just add 3 to the reward?) and in (b) reward is multiplied by a cosine? The Appendix does not clarify anything.

Minor:
- $alg_{\mathcal{E}}$ is used to denote the expert algorithm, and $\varepsilon$ to denote the approximation error, and these epsilons look very similar.
- The “base” algorithm is sometimes denoted by alg_B and sometimes by alg_E throughout. Assumption 2 has alg_B in the text and alg_E in the equation. I assume they mean the same thing - please pick one and make the text consistent.
- Appendix C.4 (proof of Thm 5) starts with “consider ${\bf x}_t^{(i)} \in \mathbb{R}^d$ " but then never references this x again

Significance:

The submission advances the understanding of in-context RL by transformers to the setting of non-stationary RL, which is important. It does not provide any new algorithms, and the architecture changes used in the proof are not evaluated empirically (perhaps a bit cumbersome to implement). Therefore this is unlikely to influence practitioners.

Originality:

The paper puts together the results of Lin et al (2024) and We and Luo (2021), and mostly looks at the expressivity of transformers when it comes to the MASTER algorithm. While this is not especially original, it is highly non-trivial to execute and seems like a worthy contribution.

---

> ### Author Rebuttal · Authors · 2025-07-31
>
> We are grateful to the reviewer for their thorough review and valuable feedback on our paper. In the following, we hope to address each point raised in the review.
>
> > Why is the definition of the non-stationarity measure different from Wei and Luo?
>
> We thank the reviewer for raising this important point. Our initial intention was to align with the learning setup in Lin et al. (2024), and thus defining the non-stationary measure as $\Delta(t) = \max_{s,a} | R_t (s,a) - R_{t+1} (s,a) |$. However, we inadvertently wrote it as $\Delta(t) = \max_{s,s',a,a'} | R_t (s,a) - R_{t+1} (s',a') |$. We have now corrected the definition to $\Delta(t) = \max_{s,a} | R_t (s,a) - R_{t+1} (s,a) |$, which is the **upper bound** of the one in Wei et al. It is worth noting that using $\Delta(t) = \max_{s,a} | R_t (s,a) - R_{t+1} (s,a) |$ instead of the definition in Wei et al. has minimal impact on the overall approximation, while offering better alignment with the setup in Lin et al.
>
> > Does using the “non-continuous” architecture complicate adapting the results of Lin et al (2024)?
>
> No, using the "non-continuous" architecture does not complicate adapting the results of Lin et al. (2024). The non-continuous transformer is essentially equivalent to conventional transformers. In the proofs of Lin et al. (2024), a specific transformer was constructed to approximate each RL algorithm, demonstrating the existence of a transformer that can achieve the error bound. Similarly, we constructed a specific transformer to approximate the MASTER algorithm. While the non-continuous transformer may differ in terms of augmented inputs, the main impact of these inputs is simply increasing the model size. Since our work is focused on the theoretical approximation rather than practical considerations, this does not complicate adapting the results of Lin et al. (2024).
>
> In fact, our initial proof used a conventional transformer without augmented inputs and relied on multiple heads to compensate. However, due to the complexity of the proof, we later switched to the augmented input strategy for simplification.
>
> We appreciate the reviewer for highlighting this, and we will include a brief explanation of the non-continuous transformer in the revised version of the main text.
>
> > Should the transformer algorithm be trained to output both the auxiliary quantity and the actions? Or should it be given access to some other function computing auxiliary values? Or does it suffice for the transformer to be able to compute these values in some intermediate block?
>
> Thank you for this insightful question. In conventional RL algorithms, auxiliary quantities (e.g., UCB scores, Q-values) are explicitly computed to inform action selection. In our transformer-based framework, we adopt a different but related strategy: the transformer is trained to output a probability distribution over actions, from which the action is sampled. This output serves a role of implicit encoding for the auxiliary information used to guide decision-making.
>
> In other words, the auxiliary quantity is not a separate supervised target, but rather emerges from the transformer’s internal computation. Our theoretical construction (see Proposition 3 and Theorem 5) shows that the transformer can implement implicit mechanisms, such as window schedulers and restart tests, that rely on internal auxiliary statistics. Thus, it is sufficient for the transformer to internally represent and use such quantities, rather than outputting them explicitly or requiring access to an external function.
>
> We appreciate the reviewer for raising this important point, and will add a brief explanation of this in the revised version to clarify how these auxiliary quantities are handled in the transformer framework.
>
> > The discussion of limitations seems inadequate. The authors should additionally discuss the discrepancies between the analyzed non-continuous transformer and the standard architecture. The limitations discussed in Lin et al (2024) also apply to this work and could be mentioned.
>
> Thank you for your suggestions. We will expand the discussion of limitations in the revised version to address the important points you raised.
>
> In particular, we will explicitly discuss the discrepancies between our analyzed non-continuous transformer and standard transformer architectures used in practice. Our theoretical construction relies on controlled masking and conditional restarts, which, while implementable in principle, do not directly correspond to standard autoregressive or encoder-decoder implementations. This mismatch may affect practical applicability and will be acknowledged clearly.
>
> We also agree that the limitations discussed in Lin et al. (2024) such as brittleness under distribution shift, reliance on synthetic pretraining distributions, and sensitivity to context length—are relevant to our setting as well, and we will incorporate them in our limitations section.
>
> We’ll also address the gap between theory and experiments, as well as the generalizability of our results to broader RL settings, since we only did experiments on bandit settings.
>
> > In the experimental setup, the first paragraph states that the action set is changing over the rounds, but the subsequent text suggests that the actions are fixed and the range of the rewards is changing. In (a) it changes to [3, 4] for some rounds (how? do you just add 3 to the reward?) and in (b) reward is multiplied by a cosine? The Appendix does not clarify anything.
>
> We apologize for the unclear explanation in the original text. To clarify:
> * In (1), the rewards in the rounds $t\in[50,100]\cup[350,400]$ are within the range $[3,4]$, while for the rest of the rounds, the reward $r\in[0,1]$. To achieve this, we added 3 to the rewards in the specified rounds.
> * In (2), the reward is indeed multiplied by a cosine function, as stated.
> We will revise the manuscript to make these details clearer and avoid any confusion in the future. Thank you for your understanding.
>
> > Clarity: In particular, the non-continuous transformer is only introduced and defined in the appendix, and the description is difficult to parse. This architecture and data augmentation should be discussed (or at least mentioned) in the main body.
>
> Thank you for the suggestion. We have added descriptions and discussion of the relevant parts in the main text to provide greater clarity.

---

> > ### Comment · Reviewer_e7t9 · 2025-08-02
> > **thanks!**
> >
> > Thanks for the clarifications! Do you think the proof could work with a regular transformer or is it a dead end?
> > It seems like the paper would require non-trivial rewriting to address some of the changes (like describing the non-continuous transformer in the main text). I'll keep the borderline accept score.

---

> ### Author Response · Authors · 2025-08-03
> **An Alternative Approximation via a Regular Transformer**
>
> We thank the reviewer for the comment. It is indeed possible to use a regular transformer, and we apologize if this wasn't clear in our previous rebuttal. While approximating MASTER with a regular transformer is certainly feasible, it would introduce substantial mathematical complexity that is orthogonal to our main contributions. We chose the augmented input approach to keep the presentation focused on our core algorithmic innovations rather than the technical details of the approximation. Below, we clarify the motivation for using augmented inputs and provide an alternative approximation approach using a regular transformer.
>
> The core challenge in using a regular transformer lies in approximating $\sigma_2$​, which must select the first non-zero entry:
>
> $$\sigma_2(\\{(s_t^{(i)}, a_t^{(i)}, r_t^{(i)})\\}_{i=0}^n) = [0; \ldots; 0; (s_t^{(k)}, a_t^{(k)}, r_t^{(k)}); 0; \ldots; 0]$$
>
> where $k = \min\\{j \in \\{0, \ldots, n\\} : (s_t^{(j)}, a_t^{(j)}, r_t^{(j)}) \neq 0\\}$. This operation requires comparing all $n+1$ trajectories simultaneously to identify $k$, and it is the main reason why the augmented inputs are used.
>
> One alternative approach is to use a regular transformer with $n+1$ heads to approximate $\sigma_1$ and $\sigma_2$:
>
> $\bullet$ Layer 1 (**Compression**): Each head uses masked multi-head attention to extract per-trajectory representations, i.e., $\mathbf{H}^{(i)}=\mathbf{W}_i^{\mathrm{comp}}\cdot \mathrm{MaskAttn}_m(\mathbf{H})\in\mathbb{R}^{\frac{D}{n+1}\times T}$. Here, $\mathrm{MaskAttn}$ restricts information flow across heads, and the combined output is $\mathbf{H}' = [\mathbf{H}^{(0)};...;\mathbf{H}^{(n)}]\in\mathbb{R}^{D\times T}$.
>
> $\bullet$ Layer 2~K: Generate Bernoulli masks, block masks, and select the first non-zero trajectory to approximate $\sigma_1$ and $\sigma_2$ (same as the procedure in the paper).
>
> $\bullet$ Final Layer (**Decompression**): Reconstruct the selected trajectory: $\mathbf{H}''=\sum_{m=0}^n\mathbf{W}_m^{\mathrm{decomp}}\cdot\mathbf{H}^{(m)}\in\mathbb{R}^{D\times T}$.
>
> However, this method adds several analytical burdens: we must ensure that $\mathbf{W}_m^{\mathrm{decomp}}\cdot \mathbf{W}_m^{\mathrm{comp}}\approx{I_D}$ to preserve information, and the analysis becomes more cumbersome.
>
> In contrast, our proposed method is a simple and transparent preprocessing step where augmented inputs are generated via a linear transformation $\mathfrak{h}_t = \mathbf{M}_t \cdot \mathbf{h}_t$. This preserves the core semantics while avoiding unnecessary architectural detours. Importantly, the resulting non-continuous transformer is **structurally equivalent** to a standard transformer with transformed inputs.
>
> As for changes to the paper, we will include a brief description of the non-continuous transformer in the main text, while keeping it concise to remain within the length limits. The discussion of limitations will be moved to the appendix. We appreciate the reviewer’s consideration and helpful feedback.

---

> > ### Comment · Reviewer_e7t9 · 2025-08-04
> > **thanks**
> >
> > Thanks again. Please do add discussion to the main text. If the main result is expressivity, changing the architecture in the appendix seems deceptive. I do appreciate that a lot of work went into this paper.

---

### Official Review · Reviewer_JYfA · 2025-06-10

**Clarity:** 3
**Significance:** 4
**Originality:** 3
**Rating:** 5
**Confidence:** 2

**Summary:**

This paper considers transformers and their use in approximating reinforcement learning algorithms. In particular, they consider whether a transformer can be trained to act as an RL algorithm when given an interaction history. Previous work (Lin et al 2024) showed that a sufficiently large transformer model can approximate popular near-optimal online RL algorithms such as linUCB. However, the question of whether transformers can adapt to non-stationary environments has not been answered. In a non-stationary environment, the reward can evolve over time. The paper proves that a transformer can achieve cumulative regret matching the minimax optimal rate for non-stationary environments.

**Questions:**

My main question is how strong of an assumption is Assumption 2 (realizability)? I am not familiar with this assumption and would like some more context surrounding it.

**Ethical Concerns:**

["NO or VERY MINOR ethics concerns only"]

**Final Justification:**

I am keeping my score. I think the paper is generally well written and I like the theoretical results. I thank the authors for providing more context regarding their assumption.

**Limitations:**

yes

**Quality:**

3

**Strengths And Weaknesses:**

Strengths:
1) The theoretical results seem strong and sound, although I did not fully verify the proofs.
2) The paper is pretty well-written and understandable.
3) There are empirical results to back up the theoretical results.

Weaknesses:
1) Assumption 2 seems a bit strong, and there is little discussion on its importance and whether it is standard/
2) There are few minor typos and grammar mistakes.

---

> ### Author Rebuttal · Authors · 2025-07-31
>
> We are grateful to the reviewer for their thorough review and valuable feedback on our paper. In the following, we hope to address each point raised in the review.
>
> **Q1/W1:**
>
> > How strong Assumption 2 is?
>
> We appreciate the reviewer raising the question. While Assumption 2 is indeed strong and difficult to verify empirically, it builds on solid theoretical foundations. For example, [1] shows that transformers are universal approximators over distributions, which supports the feasibility of such an assumption in principle.
>
> It’s also worth emphasizing that the assumption doesn't require the transformer to exactly replicate the expert policy. Rather, it assumes the transformer can learn an *amortized algorithm* whose output distribution matches that of the expert, which is a more relaxed and realistic target in many cases.
>
> A similar assumption is also used in [2], and several previous works (e.g., [3, 4, 5, 6]) have proved the expressivity of transformers from other aspects.
>
> **W2:**
>
> > There are few minor typos and grammar mistakes.
>
> Thank you for pointing this out. We have carefully revised the manuscript to correct all identified typos and grammatical issues.
>
> [1] Furuya, T., de Hoop, M. V., & Peyré, G. Transformers are Universal In-context Learners. In The Thirteenth International Conference on Learning Representations.
>
> [2] Lin, L., Bai, Y., & Mei, S. Transformers as Decision Makers: Provable In-Context Reinforcement Learning via Supervised Pretraining. In The Twelfth International Conference on Learning Representations.
>
> [3] Yun, C., Bhojanapalli, S., Rawat, A. S., Reddi, S., & Kumar, S. Are Transformers universal approximators of sequence-to-sequence functions?. In International Conference on Learning Representations.
>
> [4] Zhang, R., Frei, S., & Bartlett, P. L. (2024). Trained transformers learn linear models in-context. Journal of Machine Learning Research, 25(49), 1-55.
>
> [5] Bai, Y., Chen, F., Wang, H., Xiong, C., & Mei, S. (2023). Transformers as statisticians: Provable in-context learning with in-context algorithm selection. Advances in neural information processing systems, 36, 57125-57211.
>
> [6] Ahn, K., Cheng, X., Daneshmand, H., & Sra, S. (2023). Transformers learn to implement preconditioned gradient descent for in-context learning. Advances in Neural Information Processing Systems, 36, 45614-45650.

---

> > ### Comment · Reviewer_JYfA · 2025-08-02
> >
> > Thank you to the authors for answering my question.

---

### Official Review · Reviewer_NBt8 · 2025-06-29

**Clarity:** 3
**Significance:** 4
**Originality:** 4
**Rating:** 5
**Confidence:** 2

**Summary:**

This paper investigates the performance of transformers in non-stationary reinforcement learning environments. Building on prior work that showed transformers can approximate optimal online RL algorithms in stationary settings, this study extends the analysis to dynamic environments where reward distributions and transitions evolve over time. The authors prove that a transformer, trained in a supervised in-context learning setting, can achieve minimax optimal cumulative regret for non-stationary environments for bounded changes. They provide theoretical guarantees on generalization under distribution shifts and identify architectural requirements for transformers to implement restart-style strategies necessary for adapting to non-stationarity. A novel proof technique based on internal algorithm selection is introduced.

**Questions:**

While your theoretical results are strong for bandit settings, what are the challenges in extending this to more general RL algorithms and potentially more structured models of change?

**Ethical Concerns:**

["NO or VERY MINOR ethics concerns only"]

**Final Justification:**

Why Accept: the paper addresses an interesting and important question of understanding the behavior of transformers as in-context bandit/RL learners; elegant internal algorithm selection idea; insights for designing more robust transformer architecture; establishes matching upper and lower bounds

Why not strong accept: concerns on data efficiency remain

**Limitations:**

Yes, the authors have addressed the limitations and potential negative social impact of their work.

**Quality:**

4

**Strengths And Weaknesses:**

**Strengths:**
- The problem of understanding transformer behavior for non-stationary reinforcement learning is important
- The paper establishes a $\mathcal{\tilde{O}}(T^{2/3})$ regret bound for an in-context bandit learner matching the information theoretic lower bound
- The elegant internal algorithm selection approach offers a new method for analyzing in-context learning in transformers by conceptually formalizing how they might choose between different internal strategies
- Identifying specific architectural features that enable adaptation to non-stationarity provides valuable insights for designing more robust transformer models


**Weaknesses:**
- While the paper sets up a general RL environment, the regret optimality results and much of the discussion are heavily geared towards bandits. While bandits serve as an excellent starting point for theoretical analysis, it is not entirely clear how directly these results translate to more complex general reinforcement learning settings.
-  The reliance on a supervised in-context learning setup for the transformer raises questions about data efficiency in practical non-stationary RL. Continuously generating or providing optimal input-output pairs for training the transformer to adapt to changing dynamics might be prohibitively expensive or impossible.

---

> ### Author Rebuttal · Authors · 2025-07-31
>
> We are grateful to the reviewer for their thorough review and valuable feedback on our paper. In the following, we hope to address each point raised in the review.
>
> **Q1/W1:**
>
> > While your theoretical results are strong for bandit settings, what are the challenges in extending this to more general RL algorithms and potentially more structured models of change?
>
> Thank you for the insightful question. While our experiments are limited to bandit settings, the theoretical framework extends to more general reinforcement learning (RL) problems, as long as the expert policy can be expressed as a distribution over actions given trajectories—a property shared by many episodic or tabular RL algorithms.
>
> A key challenge in general RL is the added complexity from temporal credit assignment and state transitions. Unlike bandits, RL requires modeling long-term dependencies, which may necessitate deeper architectures and longer contexts.
>
> To address this, our approach does not require the transformer to replicate the expert’s full decision process. Instead, it *approximates the expert’s output distribution (amortized inference)*, which is more tractable and better aligned with in-context learning paradigms. This relaxation is consistent with prior work (e.g., [1]) and backed by universal approximation results.
>
> Extending our analysis to structured forms of non-stationarity (e.g., piecewise-stationary MDPs) is an important next step. We believe our decomposition, especially of restart and windowing mechanisms, provides a promising foundation, and we now discuss this in the conclusion.
>
> **W2:**
>
> > Continuously generating or providing optimal input-output pairs for training the transformer to adapt to changing dynamics might be prohibitively expensive or impossible.
>
> We appreciate the reviewer’s observation. Indeed, in certain non-stationary reinforcement learning settings, conventional RL algorithms may offer greater reliability or data efficiency than transformer-based approaches. However, the primary goal of our work is to establish a theoretical guarantee for the robustness of transformers under non-stationarity, rather than to optimize for data efficiency or other empirical performance aspects.
>
> Corollary 2 states that the regret bound holds when the number of training samples $N\geq CT^3 \log T$. If we take $T=10^3$, which already covers the horizon of many RL tasks, this gives a data requirement on the order of $10^9$. While this is certainly large compared to what's typically needed in traditional RL algorithms, it's still relatively modest compared to the scale of data used to train LLMs.
>
> [1] Furuya, T., de Hoop, M. V., & Peyré, G. Transformers are Universal In-context Learners. In The Thirteenth International Conference on Learning Representations.

---

> ### Comment · Reviewer_NBt8 · 2025-08-03
>
> Thank you for the response! This answers my questions.

---

### Official Review · Reviewer_kshj · 2025-07-03

**Clarity:** 3
**Significance:** 3
**Originality:** 3
**Rating:** 4
**Confidence:** 3

**Summary:**

This paper investigates the ability of transformer models to achieve optimal dynamic regret in non-stationary RL environments via in-context learning. The authors show that transformers, when trained with supervised learning on trajectories generated by expert algorithms, can implement mechanisms akin to restart-based and sliding-window strategies—common in adaptive RL—to adapt to non-stationarity. The main theoretical result proves that a transformer-based policy can achieve the dynamic regret bound in non-stationary RL.

**Questions:**

1. The experiments are limited to linear bandit environments. How do the authors anticipate their theoretical findings will extend to more general reinforcement learning settings, particularly those involving large or continuous state spaces and longer temporal dependencies?
2. Assumption 2 requires that the transformer can approximate the base algorithm arbitrarily well. How realistic is this in practice, especially for complex algorithms or environments with long horizons?

**Ethical Concerns:**

["NO or VERY MINOR ethics concerns only"]

**Final Justification:**

My main concerns have been addressed.

**Limitations:**

yes

**Quality:**

3

**Strengths And Weaknesses:**

**Strengths:**

1. The paper tackles an important and under-explored question: whether transformers trained via in-context learning can effectively handle non-stationary reinforcement learning environments. This is an important extension over prior work, which mostly focuses on stationary settings.
2. The authors prove that transformers can achieve optimal dynamic regret rates. The regret decomposition into approximation, training, and algorithmic terms is insightful and clearly articulated. Moreover, this paper shows that transformers can approximate key non-stationary strategies like windowing and restarts.
3. The empirical results show that transformers can match or exceed baseline and expert algorithms, even in previously unseen conditions.

**Weakness:**:

1. The theoretical guarantees hinge on a realizability assumption: that the transformer can approximate the expert policy to arbitrary precision. Although motivated by prior work, this assumption is strong and not easily verifiable in practice.
2. The architectural implementation of non-stationary mechanisms (e.g., the restart mechanism via 2-layer MLPs) is elegant, but it may oversimplify real-world scenarios. The paper does not explore how robust these designs are to scaling or how sensitive performance is to architecture choices.
3. While the experiments are well designed, they are limited to linear bandit environments. The results may not generalize to more complex non-stationary MDPs or higher-dimensional, structured RL tasks. This limits the empirical reach of the theoretical claims.

---

> ### Author Rebuttal · Authors · 2025-07-31
>
> We are grateful to the reviewer for their thorough review and valuable feedback on our paper. In the following, we hope to address each point raised in the review.
>
> **Q2/W1:**
>
> > Assumption 2 requires that the transformer can approximate the base algorithm arbitrarily well. How realistic is this in practice, especially for complex algorithms or environments with long horizons?
>
> Thank you for raising this important point regarding the realizability assumption. We clarify below its motivation, scope, and practical implications.
>
> **Theoretical grounding:** Assumption 2 is based on prior work showing that transformers can emulate standard reinforcement learning algorithms such as LinUCB and Thompson Sampling via in-context learning [1]. Furthermore, [2] shows that transformers are universal approximators over distributions, which supports the feasibility of such an assumption in principle. It’s also worth emphasizing that the assumption doesn't require the transformer to exactly replicate the expert policy. Rather, it assumes the transformer can learn an *amortized algorithm* that reproduces the expert’s output distribution, which is a more relaxed and realistic target in many cases.
>
> **Practical feasibility:** We emphasize that Assumption 2 does not imply that any complex algorithm or environment is immediately approximable by arbitrary transformers. Rather, it reflects a scaling law—namely, that approximation error $\varepsilon$ can be made small by increasing model size and training data, as reflected in the regret bound (Theorem 1) and its dependency on architecture parameters (e.g., $D' = O(\sqrt{T \log^2 T / \varepsilon})$). In this sense, the assumption is realistic to the extent that the base algorithm itself is implementable by a recurrent or sequential decision process. For many commonly studied algorithms, including sliding-window UCB and restart-based methods like MASTER, this has been shown to be the case in structured architectures [3].
>
> **Constructive verification:** In Section 4.2 and Appendix D.2, we construct specific transformer blocks that replicate components of non-stationary algorithms—such as window schedulers and restart logic—using standard attention and MLP modules. This shows the assumption is not merely abstract but realizable within standard architectures.
>
> **W2:**
>
> > The architectural implementation of non-stationary mechanisms (e.g., the restart mechanism via 2-layer MLPs) is elegant, but it may oversimplify real-world scenarios. The paper does not explore how robust these designs are to scaling or how sensitive performance is to architecture choices.
>
> **Theory:** Instead of exhaustively exploring design choices, our algorithmic construction is designed to demonstrate the existence of a transformer that can approximate the expert algorithm across the entire input space. For the restart mechanism specifically, many RL algorithms adopt either deterministic strategies (e.g., periodic restarts) or rule-based ones (e.g., resets triggered by specific conditions). Deterministic approaches can be viewed as a form of window scheduling, where the algorithm either discards historical data or resets its internal state at fixed intervals. Rule-based strategies often rely on binary indicators, which can be naturally modeled by a 2-layer MLP. In this sense, our architectural implementation of non-stationary mechanisms is expressive enough to capture a broad class of restart behaviors, at least from a theoretical standpoint.
>
> **Experiment:** While more in-depth experiments on scaling and performance sensitivity of the architecture are left to future work, we conducted experiments using a transformer with 4 heads and 4 layers (compared to the 16 heads and 16 layers described in the paper) under the Low Non-stationarity setting to address the reviewer’s concerns. The "Regret Over Time" results are summarized in the table below. It demonstrates that the transformer is capable of achieving performance comparable to that of the expert, even with 4 heads and 4 layers.
>
> | Model\Round | 50 | 100 | 200 | 300 | 400 | 500 | 600 | 700 | 800 | 900 | 1000 |
> |-------------|-----|-----|-----|-----|-----|-----|-----|-----|-----|-----|------|
> | Transformer | 14.5 | 29.5 | 59.7 | 89.8 | 119.9 | 150.1 | 175.3 | 205.6 | 235.5 | 269.2 | 301.2 |
> | LinUCB | 13.3 | 26.1 | 51.7 | 77.3 | 102.8 | 128.3 | 153.7 | 179.2 | 210.3 | 235.7 | 262.3 |
> | LinUCB+MASTER | 14.3 | 29.0 | 57.1 | 83.7 | 111.0 | 138.8 | 165.5 | 193.7 | 221.8 | 250.1 | 278.3 |
>
> **Q1/W3:**
>
> > How do the authors anticipate their theoretical findings will extend to more general reinforcement learning settings, particularly those involving large or continuous state spaces and longer temporal dependencies?
>
> Thank you for the question. We believe that the implications from our results are applicable to settings such as continuous state spaces and longer temporal dependencies. As this work is primarily theoretical, we initially focused less on experiments. However, to better address the reviewer’s concerns, we extended our evaluation to a continuous state space environment.
>
> We ran additional experiments on the MuJoCo Hopper task, which features an 11D continuous state space (e.g., joint angles, velocities) and a 3D continuous action space corresponding to joint torques. The objective is to learn a hopping behavior that balances forward progress, energy efficiency, and stability. Rewards are shaped accordingly to encourage forward velocity while penalizing excessive control effort and contact forces.
>
> We used Soft Actor-Critic (SAC) [4] as our base algorithm and SAC+MASTER as the expert. For comparison, we trained a 3-layer transformer (with 256D embeddings) on stationary data for 20,000 episodes. During training, the transformer was evaluated on non-stationary data every 100 episodes, by switching to ```model.eval()``` to perform evaluation for 10 episodes, and then switching back to ```model.train()``` for further training. Similarly, SAC was trained on stationary data, and every 10 episodes, it was evaluated in a non-stationary environment.
>
> This setup follows prior work (e.g., [5, 6]). Although individual state components range over $\mathbb{R}$, typical ranges are more bounded: the z-axis lies roughly within [0, 2], joint angles within [–1, 1], and velocities within [–10, 10]. We introduce non-stationarity by adding a drift value to the states, defined as $\sin(0.01 t)$. (new_state = state + $\sin(0.01 t)$)
>
> **Results:**
>
> **Transformer:**
>
> | Episode | 100 | 200 | 300 | 400 | 500 | 1K | 1K~10K | 10K~20K |
> |---------|-----|-----|-----|-----|-----|-----|--------|----------|
> | Avg. Reward | 113 | 54 | 68 | 52 | 51 | 61 | 42~62 | 42~47 |
>
> **SAC:**
>
> | Episode | 200 | 400 | 1K | 2K | 2K~5K | 5K~500K |
> |---------|-----|-----|----|----|-------|--------|
> | Avg. Reward | 26 | 42 | 40 | 40 | 6~40 | 5~44 |
>
> **SAC+MASTER:**
>
> | Step | 200 | 400 | 1K | 2K | 2K-5K | 5K-50K | 50K-100K | 100K-1M |
> |------|-----|-----|----|----|-------|--------|----------|---------|
> | Avg. Reward | 14 | 12 | 19 | 30 | 10-214 (mostly 10-30) | 9-201 (mostly 10-30) | 7-180 (mostly 10-30) | 6-217 (mostly 10-30) |
>
> ** Typically, SAC requires over 1M steps to converge. Due to time constraints, we only trained SAC+MASTER for 1M steps, as its learning process is much faster than SAC's due to the restart mechanism, which maintains smaller active memory pools and results in faster computation.
>
> ** SAC+MASTER Avg Reward after 900K steps (converged at this phase): Among all rewards in the evaluation episodes, 21% of the rewards are over 50, 13% over 70, 7% over 100, and 2% over 150.
>
> The results show that the transformer maintains more stable performance under observation drift and achieves rewards comparable to the expert. While the reported SAC and transformer scores may appear lower than those in earlier work (which often exceed 1,000), we validated our SAC and transformer implementation under standard stationary conditions and confirmed it aligns with previous findings before introducing non-stationarity.
>
> We acknowledge that this single environment does not fully capture the complexity of general RL tasks, but we view this as a meaningful first step toward evaluating the theoretical framework in richer, more dynamic environments. We plan to explore additional tasks with longer horizons and more complex dynamics in future work.
>
> [1] Lin, L., Bai, Y., & Mei, S. Transformers as Decision Makers: Provable In-Context Reinforcement Learning via Supervised Pretraining. In The Twelfth International Conference on Learning Representations.
>
> [2] Furuya, T., de Hoop, M. V., & Peyré, G. Transformers are Universal In-context Learners. In The Thirteenth International Conference on Learning Representations.
>
> [3] Wei & Luo (2021). Non-stationary reinforcement learning without prior knowledge: An optimal black-box approach. COLT.
> [4] Haarnoja, T., Zhou, A., Abbeel, P., & Levine, S. (2018, July). Soft actor-critic: Off-policy maximum entropy deep reinforcement learning with a stochastic actor. In International conference on machine learning (pp. 1861-1870). Pmlr.
>
> [5] Chen, L., Lu, K., Rajeswaran, A., Lee, K., Grover, A., Laskin, M., ... & Mordatch, I. (2021). Decision transformer: Reinforcement learning via sequence modeling. Advances in neural information processing systems, 34, 15084-15097.
>
> [6] Huang, S., Hu, J., Yang, Z., Yang, L., Luo, T., Chen, H., ... & Yang, B. (2024). Decision mamba: Reinforcement learning via hybrid selective sequence modeling. Advances in Neural Information Processing Systems, 37, 72688-72709.

---

> > ### Comment · Reviewer_kshj · 2025-08-04
> >
> > Thank you for your responses in addressing my concerns. Happy to raise my rating to 4.

---

### Note · Authors · 2025-08-13

We sincerely thank all reviewers for their valuable feedback and encouraging remarks. We are pleased that our contributions were well-received, and we greatly appreciate the recognition of the strengths of our work, including:
* Addressing an important and under-explored question: whether transformers trained via in-context learning can effectively handle non-stationary reinforcement learning environments (Reviewers kshj, NBt8).
* Making a non-trivial contribution with strong and sound theoretical results (Reviewers JYfA, e7t9).
* Presenting elegant and well-articulated proofs (Reviewers kshj, NBt8).

**Addressing Reviewer Concerns**

A central concern across reviews was that **Assumption 2** might be too strong. In our rebuttal, we addressed this by providing theoretical grounding from prior work, discussing its practical feasibility for complex algorithms and long horizons with constructive verification, and explaining that the assumption targets an amortized reproduction of the expert’s output distribution rather than exact policy replication. This clarified why the assumption is not as restrictive as it may seem and resolved this concern.

Reviewer kshj noted that our experiments were limited to linear bandit environments and questioned generalization to more complex RL tasks. We addressed this by adding experiments in a continuous state-space environment, finding results consistent with those reported in the paper.

Reviewer e7t9 expressed concern that using a non-continuous transformer in our proof might weaken its validity. We have now shown that the proof also holds for a regular transformer, and clarified that the non-continuous version was used solely for proof simplicity and is structurally equivalent to the regular transformer.

**Changes in the Revised Manuscript**

We will make minor clarifications to improve clarity and presentation, including moving the explanation of the non-continuous transformer from the appendix to the main text (with detailed motivation and proof retained in the appendix), slightly expanding the appendix discussion of Assumption 2 with relevant theoretical background, and adding a brief note on limitations.

We are grateful that the reviewers recognize our work’s potential to inspire further research in the community, and we look forward to presenting a clearer and more refined version of our paper.

---

### Decision · Program_Chairs · 2025-09-17

**Decision:**

Accept (poster)

**Comment:**

This paper demonstrates that transformers can achieve optimal dynamic regret using in-context learning in non-stationary RL environments.  After discussion, the reviewers were uniformly positive about the paper.  There were some concerns about the strength of the assumptions (e.g., the realizability assumption) that I read as having been resolved in discussion.  The initial experimental evaluation was limited to a bandit setting; in response to reviewer comments, the authors have added an evaluation on a continuous MuJoCo environment (the Hopper task).

Overall this seems like a sound contribution that is likely to be influential.